# Validation of the IASI FORLI/Eumetsat ozone products using satellite (GOME-2), ground-based (Brewer-Dobson, SAOZ, FTIR) and ozonesonde measurements

Anne Boynard[1,2], Daniel Hurtmans[3], Katerina Garane[4], Florence Goutail[1], Juliette Hadji-Lazaro[1], Maria Elissavet Koukouli[4], Catherine Wespes[3], Corinne Vigouroux[5], Arno Keppens[5], Jean-Pierre Pommereau[1], Andrea Pazmino[1], Dimitris Balis[4], Diego Loyola[6], Pieter Valks[6], Ralf Sussmann[7], Dan Male[8], Pierre-François Coheur[3], and Cathy Clerbaux[1,3]

[1]LATMOS/IPSL, Sorbonne Universités, UVSQ, CNRS, Paris, 75252, France
[2]SPASCIA, Ramonville-Saint-Agne, 31520, France
[3]Université libre de Bruxelles, Atmospheric Spectroscopy, Service de Chimie Quantique et Photophysique, Brussels, 1050, Belgium
[4]Laboratory of Atmospheric Physics, Aristotle University of Thessaloniki, Thessaloniki, 54124, Greece
[5]Royal Belgian Institute for Space Aeronomy (BIRA-IASB), Brussels, 1180, Belgium
[6]Institut für Methodik der Fernerkundung (IMF), Deutsches Zentrum für Luft- und Raumfahrt (DLR), Oberpfaffenhofen, Germany
[6]Karlsruhe Institute of Technology, IMK-IFU, Garmisch-Partenkirchen, Germany
[7]National Institute of Water and Atmospheric Research Ltd (NIWA), Lauder, New Zealand

*Correspondence to*: Anne Boynard (anne.boynard@latmos.ipsl.fr)

**Abstract.** This paper assesses the quality of IASI/Metop-A (IASI-A) and IASI/Metop-B (IASI-B) ozone ($O_3$) products (total and partial $O_3$ columns) retrieved with the Fast Optimal Retrievals on Layers for IASI Ozone (FORLI-$O_3$) v20151001 software for nine years (2008 – July 2017) through an extensive inter-comparison and validation exercise using independent observations (satellite, ground-based and ozonesonde). Compared to the previous version of FORLI-O3 (v20140922), several improvements were introduced in FORLI-$O_3$ v20151001, including absorbance look-up tables recalculated to cover a larger spectral range, with additional numerical corrections. This leads to a change of ~4% in the Total Ozone Column (TOC) product, which is mainly associated with a decrease in the retrieved $O_3$ concentration in the middle stratosphere (above 30 hPa/25 km). IASI-A and IASI-B TOCs are consistent, with a global mean difference less than 0.3% for both day- and nighttime measurements, IASI-A being slightly higher than IASI-B. A global difference less than 2.4 % is found for the tropospheric (TROPO) $O_3$ column product (IASI-A being lower than IASI-B), which is partly due to a temporary issue related to IASI-A viewing angle in 2015. Our validation shows that IASI-A and IASI-B TOCs are consistent with GOME-2, Dobson, Brewer, SAOZ and FTIR TOCs, with global mean differences in the range 0.1 – 2 % depending on the compared instruments. The worst agreement with UV-vis retrieved TOC [satellite and ground] is found at the southern high latitudes. The IASI-A and ground-based TOC comparison for the period 2008 – July 2017 shows long-term stability of IASI-A, with insignificant or small negative drift among 1 – 3 % decade[-1]. The comparison results between IASI-A and IASI-B against smoothed FTIR and ozonesonde partial $O_3$ columns vary in altitude and latitude, with maximum standard deviation for the 300-150 hPa column (20-40 %) due to strong ozone variability and large total retrievals errors. Compared to ozonesonde data, IASI-A and IASI-B

O$_3$ TROPO column (defined as the column between the surface and 300 hPa) is positively biased in the high latitudes (4-5 %) and negatively biased in the mid-latitudes and tropics (11-13 % and 16-19 %, respectively). The IASI-A-to-ozonesonde TROPO comparison for the period 2008 – 2016 shows a significant negative drift in the Northern Hemisphere of -8.6±3.4 % decade$^{-1}$, which is also found in the IASI-A-to-FTIR TROPO comparison. When considering the period 2011 – 2016, the drift

value for the TROPO column decrease and become statistically insignificant. The observed negative drifts of IASI-A TROPO O$_3$ product (8-16% decade$^{-1}$) over 2008-2017 might be taken into consideration when deriving trends from this product and this time period.

## 1 Introduction

Ozone (O$_3$) plays a major role in the chemical and thermal balance of the atmosphere. In the stratosphere, O$_3$ protects the

biosphere and humans from harmful ultraviolet (UV) radiation. In the troposphere, O$_3$ plays different roles depending on the altitude region. Near the surface, ozone in excessive amounts is one of the main air pollutants impacting both human health (Brunekreef and Holgate, 2002; Lim et al., 2012) and ecosystems (Fowler et al., 2009). In the upper troposphere, ozone is an important anthropogenic greenhouse gas (IPCC, 2013) and acts as a short-lived climate forcer (Shindell et al., 2012). Tropospheric O$_3$ originates either from complex photochemical reactions involving nitrogen oxides (NO$_x$), carbon monoxide

(CO) and hydrocarbons (e.g. Chameides and Walker, 1973; Crutzen, 1973) or from the stratosphere by downward transport to the troposphere especially at mid- and high latitudes (e.g. Holton et al., 1995) as well as from long-range transport (e.g., Stohl and Trickl, 1999). The lifetime of tropospheric ozone varies with altitude and ranges from 1-2 days in the boundary layer where dry deposition is the major sink to several weeks in the free troposphere, so that the transport scale of O$_3$ can be intercontinental and hemispheric (Monks et al., 2015). To better understand its variability and impacts, it is therefore crucial

to obtain information on its vertical, spatial and temporal distribution. These can be provided by observations from space-borne instruments.

The Infrared Atmospheric Sounding Interferometer (IASI) is a nadir-viewing spectrometer flying on board the Eumetsat's (European Organisation for the Exploitation of Meteorological Satellites) Metop-A and Metop-B satellites, since October 2006 and September 2012 respectively. In order to ensure the continuity of IASI observations for atmospheric composition

monitoring, a third satellite (Metop-C) is scheduled to be launched in September 2018. Thanks to the nadir geometry complemented by off-nadir measurements up to 48.3° on both sides of the satellite track (swath of about 2200 km), each IASI instrument covers the globe twice a day, with a field of view of 4 pixels of 12 km in diameter on the ground at nadir. The two Metop satellites are on the same orbit with Equator crossing times of 09:30 (21:30) local mean time solar time for the descending (ascending) part of the orbit. There are therefore numerous common observations between two consecutive tracks.

However, since Metop-A and Metop-B are 180° out of phase, there is a ~50 min temporal difference between both instruments (one satellite might be before or after the other); thus the observations are never quite simultaneous. In addition, the geometry

of the observations is different and generally off-nadir with opposite angles, so the location of the observation between the two instruments varies and thus the pixels are not absolutely geographically co-localized.

Having a twice daily coverage and a 12-km diameter footprint at nadir, IASI has the potential for providing measurements for $O_3$ globally, with a high spatial resolution. Previous studies have demonstrated the ability of IASI to measure $O_3$ separately in the stratosphere (Scannell et al., 2012; Gazeaux et al., 2013), in the upper troposphere and lower stratosphere (UTLS) (e.g. Barret et al., 2011; Wespes et al., 2016), and in the troposphere (e.g. Eremenko et al., 2008; Dufour et al., 2010, 2015; Safieddine et al., 2013, 2014). Using the long-term IASI $O_3$ record, interannual variability of tropospheric ozone and long-term trends can be derived (Safieddine et al., 2016; Wespes et al., 2016; 2018; Gaudel et al., 2018). Lately, Wespes et al. (2017) analyzed more than eight years of IASI $O_3$ data to identify the main geophysical drivers (e.g., solar flux, the Quasi-Biennial Oscillation, North Atlantic Oscillation, El Niño-Southern Oscillation) of $O_3$ regional and temporal variability.

Several research groups have developed $O_3$ retrieval algorithms for IASI based on different approaches (e.g. Barret et al., 2011; Dufour et al., 2012; Hurtmans et al., 2012; Oetjen et al., 2016). In particular, ULB & LATMOS have developed the Fast Optimal Retrievals on Layers for IASI $O_3$ (FORLI- $O_3$) software (Hurtmans et al., 2012), which uses the IASI Level-1C data to retrieve Level-2 $O_3$ products. A series of validation exercises of IASI $O_3$ products retrieved from different versions of FORLI- $O_3$ (v20100825, v20140922), focusing on a particular region and/or relatively short period of time were undertaken (e.g. Dufour et al., 2012; Pommier et al., 2012; Scannell et al., 2012; Gazeaux et al., 2013; Safieddine et al., 2016). Boynard et al. (2016) performed an extensive validation of IASI $O_3$ products retrieved from FORLI-$O_3$ v20140922 against a series of independent observations, on the global scale, for the period 2008 – 2014. This study reported that, on average, FORLI-$O_3$ v20140922 overestimates the ultraviolet (UV) Total Ozone Column (TOC) by 2-7% with the largest differences found at high latitudes. It is worth mentioning that Boynard et al. (2016) did not perform any comparison with measurements in other spectral ranges than UV. The comparison with ozonesonde vertical profiles shows that on average FORLI-$O_3$ v20140922 underestimates $O_3$ by ~5-15% in the troposphere while it overestimates $O_3$ by ~10-40% in the stratosphere depending on the latitude.

Several algorithm improvements were introduced later in FORLI-$O_3$, including absorbance look-up tables recalculated to cover a larger spectral range using the 2012 HITRAN spectroscopic database (Rothman et al., 2013), with additional numerical corrections. Boynard et al. (2016) evaluated 12 days of the new IASI $O_3$ products retrieved from FORLI v20151001 and found a correction of ~4% for the TOC positive bias when compared to the UV ground-based and satellite observations, bringing the overall global comparison to ~1-2% on average. It was shown that this improvement is mainly associated with a decrease in the retrieved $O_3$ concentration in the middle stratosphere (MS, above 30 hPa/25 km). This $O_3$ retrieval algorithm (FORLI-$O_3$ v20151001) is currently being implemented into the Eumetsat processing facility under the auspices of the Ozone and Atmospheric Composition Monitoring Satellite Application Facility (AC SAF) project in order to operationally distribute Level-2 IASI $O_3$ profiles to users through the EumetCast system in 2018. IASI Level-2 and Level-3 $O_3$ products processed with FORLI v20151001 are part of the European Space Agency $O_3$ Climate Change Initiative (Ozone_cci, www.esa-ozone-cci.org) and the European Centre for Medium-Range Weather Forecasts (ECMWF) Copernicus Climate Change (C3S)

projects, respectively, which focus on building consolidated climate-relevant ozone data sets as essential climate variables (ECVs). Therefore, validating the latest version of the IASI $O_3$ products over a long-time period and assessing their stability are necessary for decadal trend studies, model simulation evaluation and data assimilation applications. This is one of the main motivations of the present work. The goals of the Ozone_cci project are described in Garane et al. (2018) and its requirements in term of satellite product stability, which is defined to $1 - 3$ % / decade based on the requirements formulated by the Global Climate Observing System (GCOS) and the Climate Modeling User Group (CMUG) climate modelling community for ozone is detailed in Van Weele et al. (2016).

In this paper, we assess the quality of the IASI $O_3$ products retrieved using FORLI-$O_3$ v20151001 (hereafter referred as to "IASI $O_3$ products"), with GOME-2 also on Metop, ground-based (GB) network data (Brewer, Dobson, SAOZ and FTIR) and ozonesonde measurements. Sections 2 and 3 describe the characteristics of the datasets used for the validation and the comparison methodology, respectively. Section 4 presents the intercomparaison between IASI-A and IASI-B $O_3$ derived total and tropospheric columns. Section 5 provides the IASI-A and IASI-B TOC and partial ozone column product validation results using independent satellite, GB and ozonesonde observations. Finally, Section 6 summarizes the results from this new validation.

## 2 IASI measurements and independent datasets used for the validation

### 2.1 IASI Ozone retrievals

IASI ozone retrievals are performed in the $1025 - 1075$ cm$^{-1}$ spectral range using the optimal estimation method (OEM) (Rodgers, 2000) and tabulated absorption cross-sections at various pressures and temperatures to speed up the radiative transfer calculation. The ozone climatology by McPeters et al. (2007) is used as *a priori* information consisting in one single $O_3$ *a priori* profile and variance-covariance matrix. The Eumetsat Level-2 data (pressure, water vapor, temperature and clouds) are used as input in FORLI. It is worth mentioning that the Eumetsat dataset is not homogenous since it has been processed using different versions of the IASI Level-2 Product Processing Facility between 2008 (v4.2) and 2016 (v6.2), as summarized in Van Damme et al. (2017). The error budget of the retrieved $O_3$ profile shows that the dominant errors originate from the limited vertical sensitivity, from the measurement noise and from uncertainties in the fitted (water vapor column) or fixed (e.g. surface emissivity, temperature profile) parameters (Hurtmans et al., 2012). In order to avoid cloud contaminated scenes, retrievals are only performed for clear or almost-clear scenes with a fractional cloud cover below 13%, identified using the cloud information from the Eumetsat operational processing (August et al., 2012). In addition, no retrieval is performed for pixels characterized by an error related to the Level-1C IASI data, by no Level-2 Eumetsat data associated with Level-1C data or by missing temperature, water vapor, surface pressure or cloud value in Level-2 Eumetsat data.

The IASI $O_3$ dataset used in this paper covers the period January 2008 – July 2017. The $O_3$ product is a vertical profile given as partial columns in molecules $cm^{-2}$ in 40 layers between the surface and 40 km, with an extra layer from 40 km to the top of the atmosphere. It also includes other relevant information such as quality flags, *a priori* profile, total error profile and the averaging kernel matrix, on the same vertical grid. The following quality flags were applied to filter the dataset for further validation analysis. Specifically the data were excluded when: (i) the spectral fit residual root mean square error (RMS) is higher than $3.5 \times 10^{-8}$ W/($cm^2$ sr $cm^{-1}$), reflecting a too large difference between observed and simulated radiances; (ii) the spectral fit residual bias is lower than $-0.75 \times 10^{-9}$ W/($cm^2$ sr $cm^{-1}$) or higher than $1.25 \times 10^{-9}$ W/($cm^2$ sr $cm^{-1}$); (iii) the partial $O_3$ column is negative; (iv) there were abnormal averaging kernel values; (v) the spectral fit diverged; and (vi) the total error covariance matrix is ill conditioned; (vii) the $O_3$ profiles have an unrealistic C-shape (i.e. abnormal increase in $O_3$ at the surface, e.g. over desert due to emissivity issue), with a ratio of the surface – 6 km column to the total column higher or equal to 0.085 and (viii) the DOFS is lower than 2, which are mostly associated with bad quality data in the Antarctic region.

A representative IASI-A averaging kernel matrix is illustrated in Fig. 1a, showing the difficulty to distinguish the ozone structures between one level from another. However, it shows the altitude ranges characterized by peaks of sensitivity: ~5, 12, 18 and 40 km. Another way to visualize the AK matrix is to represent the AK profiles as a function of altitude as shown in Fig. 1b. The AK are not maximal at their nominal altitudes, which indicates that other altitudes contribute to ozone value at individual retrieval altitude. A way to estimate the vertical resolution of IASI $O_3$ profiles is to analyze the DOFS as a function of altitude. The cumulative DOFS, which is presented in Fig. 1c, is continuously increasing with altitude, given that there exists information content in the observation for the entire altitude range.

The IASI retrieval error on the TOC, including the smoothing and the measurement error, is usually below 2 %, except in the Antarctic (> 4 %), which is due to the particularly weak signal in this region. For the surface-300 hPa, 300-150 hPa, 150-25 hPa and 25-3 hPa partial columns, it is estimated to ~15 %, 17 %, 4 % and 3 % respectively.

## 2.2 The Global Ozone Monitoring Experiment-2 (GOME-2) data

The GOME-2 instrument, also on board the Metop-A and B platforms is a UV-vis-NIR (visible-near IR) nadir viewing scanning spectrometer, with an across-track scan time of 6 s and a nominal swath width of 1920 km, providing global coverage of the sunlit part of the atmosphere almost within 1.5 days (Hassinen et al., 2016; Munro et al., 2016). GOME-2 ground pixels have a footprint size of 80 km x 40 km, which is larger than that of IASI (pixel of 12 km diameter). In the framework of the Eumetsat AC SAF project, GOME-2 total ozone data are processed at DLR operationally, both in near-real time and offline, using the GOME Data Processor (GDP) algorithm (Loyola et al., 2011; Hao et al., 2014; Valks et al., 2014). The GOME-2 products has been validated using ground-based measurements (e.g Loyola et al., 2011; Koukouli et al., 2012, 2015; Hao et al., 2014), which has shown an overall agreement within 1% in most situations. As shown in Hao et al. (2014), there is an excellent agreement between the GOME-2A and GOME-2B TOCs, with a mean difference of around 0.5%. Therefore in this study, the IASI-A and IASI-B validation is limited to the comparison with GOME-2A TOC products only. In this comparison,

we only use GOME-2A TOC data meeting the valid conditions given in Valks et al. (2017): TOC value ranging between 75 and 700 Dobson units (DU) and slant column error low than 2 %.

## 2.3 Ground-based data

Daily TOC measurements from Dobson and Brewer UV spectrophotometers available for the period 2008 – 2017 were
downloaded from the World Ozone and Ultraviolet Radiation Data Centre (WOUDC, http://woudc.org). The GB stations considered in this paper (see Table A1 in Boynard et al. (2016) for a complete list of the stations) have been extensively used in a series of validation papers of satellite TOC measurements (e.g. Weber et al., 2005; Balis et al., 2007a, 2007b; Koukouli et al., 2012, 2015; Boynard et al., 2016). For the validation of IASI-A and IASI-B TOCs, only direct sun observations are used as GB UV reference data as they are the most reliable for both the Dobson and the Brewer spectrophotometers, the latter
offering an accuracy of about 1 % at moderate solar zenith angles (e.g. Kerr et al., 2002).

TOC measurements are also obtained from SAOZ (Système d'Analyse par Observation Zénithale; Pommereau and Goutail, 1988) zenith sky UV-vis spectrometers, which are part of the Network for the Detection of Atmospheric Composition Change (NDACC, (http://www.ndacc.org). The SAOZ TOC measurements are performed in the visible Chappuis bands between 450 and 550 nm with a medium spectral resolution of 1 nm, twice a day during twilight (sunrise and sunset) at solar zenith angle
ranging between 86 and 91°. The retrieval is based on the Differential Optical Absorption Spectroscopy (DOAS) procedure (Platt, 1988). Since observations are performed at twilight, SAOZ can be operated during the whole year at all latitudes up to ±67°. At latitudes higher than the polar circle, there is no measurement during permanent night in winter and during permanent day in summer. SAOZ performances have been continuously assessed by regular comparisons with UV-vis independent observations (e.g. Hofmann et al., 1995; Roscoe et al., 1999; Hendrick et al., 2011). The SAOZ total accuracy, including a 3 %
cross-section uncertainties, is ~6% (Hendrick et al., 2011). In this study, eight SAOZ stations deployed at all latitudes from the Arctic to the Antarctic (see Table 3 in Boynard et al. (2016) for their locations) are used for IASI-A and IASI-B TOC validation.

Regular ozone measurements from high-resolution solar absorption spectra recorded by GB FTIR (Fourier transform infrared) spectrometers available for the period 2008 – 2017 were downloaded from NDACC. The ozone FTIR retrieval principle, which
is based on the optimal estimation method (Rodgers, 2000), as for FORLI, is detailed in Vigouroux et al. (2008). Such measurements have the advantage to provide not only TOCs with a precision of 2 %, but also low vertical resolution profiles with about four independent partial columns, one in the troposphere and three in the stratosphere up to about 45 km, with a precision of about 5-6 % (Vigouroux et al., 2015). Therefore, the FTIR measurements are used to validate not only IASI TOCs but also IASI partial ozone columns. The stations considered in the present work were used in several papers for trend analyses
(Vigouroux et al., 2008, 2015; García et al., 2012; Wespes et al., 2016) and validation studies (Dupuy et al., 2009; Viatte et al., 2011). The latitudinal coverage ranges from 67.8°N to 45°S, so only the southern high latitudes are not covered. The location of the six FTIR stations used in the comparison is given in Table 1 and presented in Fig. 2. Since these solar absorption

measurements requires daylight conditions, there is no measurement at Kiruna during polar winter. All stations use the high-resolution spectrometers Bruker, which can achieve a resolution of 0.0035 cm$^{-1}$ or better. Details on the harmonized retrieval parameters can be found in Vigouroux et al. (2015). For all stations, the 10μm spectral region is fitted to retrieved $O_3$ using two retrieval algorithms: either PROFFIT9 at Kiruna and Izaña or SFIT2/4 at the other stations. The two algorithms have been compared in Hase et al. (2004). The spectroscopic database used is HITRAN 2008 (Rothman et al., 2009). Each station is using the daily pressure and temperature profiles from NCEP (National Centers for Environmental Prediction) and has one *a priori* profile, which is obtained from the same model WACCM4 (Whole Atmosphere Community Climate Mode; Garcia et al., 2007).

## 2.4 Ozonesonde data

High resolution ozone vertical profiles measured from ozonesonde for the period 2008 – 2017 were downloaded from the WOUDC and NOAA-ESRL (http://www.esrl.noaa.gov/gmd/dv/ftpdata.html) archives. The sondes provide measurements of $O_3$ up to 30-35 km with a vertical resolution of ~150 m. Only sonde measurements based on electrochemical concentration cells (ECCs), which measure the oxidation of a potassium iodine (KI) solution by $O_3$ (Komhyr et al., 1995), are used in this study. Their accuracy is generally good (±3-5 %) and their uncertainties are of about 10 % throughout most of the profile below 28 km (Deshler et al., 2008; Smit et al., 2007), while other types of ozonesondes have somewhat poorer accuracy (5-10%), (e.g. Hassler et al., 2014; Liu et al., 2013). A total of 56 ozonesonde stations in mid-latitudes, polar and tropical regions are considered in the present study. The location of the ozonesonde stations used in the comparison is presented in Fig. 2.

## 3. Comparison methodology

Since the characteristics are not the same from one dataset to the other, different comparison methodologies and collocation criteria are applied and described in this section. For all datasets, the differences are calculated as: [IASI – DATA] (in DU) or [100 x (IASI – DATA) / DATA] (in percent (%)), where DATA corresponds to the independent data used for the validation of IASI ozone data (ie GOME-2, Brewer/Dobson, SAOZ, FTIR and sonde ozone data).

The IASI-A and IASI-B $O_3$ products are assessed in terms of TOCs and partial ozone columns. The validation exercise is performed using the same partial columns as those used in Wespes et al. (2016) since these columns contain around one piece of information, have maximum sensitivity approximately in the middle of each of the layers, and reproduce the well-known cycles related to chemical and dynamical processes characterizing these layers: surface-300 hPa (TROPO), 300-150 hPa (UTLS), 150-25 hPa (LMS for lower and middle stratosphere) and 25-3 hPa (MS). On average, these pressure columns correspond to the following altitude columns: surface-8 km, 8-15 km, 15-22 km 22-40 km, respectively. Note, however, that for the comparison between IASI and ozonesonde data, the MS is limited to the column 25-10 hPa as sonde generally burst around 30-35 km (see Section 3.2 below). For the assessment of IASI vertical profiles, we refer to Keppens et al. (2018, this issue).

The comparison between IASI-A and IASI-B against DATA is performed over the period 2008 – 2017 and 2013 – 2017, respectively.

## 3.1 Direct comparison with GOME-2, Brewer, Dobson and SAOZ data

Since only the TOCs are provided in the independent GOME-2A, Brewer, Dobson and SAOZ datasets, a direct IASI/DATA comparison is performed in this validation exercise.

The comparison between IASI against GOME-2A TOCs is not straightforward because the pixels are not co-localized in time and space, and IASI and GOME-2 instruments have different pixel size. In order to compare collocated data, a simple way is to calculate the daily average of IASI-A, IASI-B and GOME-2A TOCs along with their relative difference over a constant 1°x1° grid cell. As the UV-vis instrument provides daytime observations, only the IASI daytime data (SZA < 90°) are used in this comparison.

For the comparison between IASI against Brewer and Dobson TOCs, the coincidence criteria are set to a 50-km search radius between the satellite pixel centre and the geocolocation of the ground-based station as well as to the same day of observations. For each GB measurement, only the closest IASI measurements are kept for the comparison.

For the comparison between IASI against SAOZ TOCs, sunrise (sunset) SAOZ measurements are compared to collocated daytime (nighttime) IASI daily data averaged in a 300 km diameter semi-circular area located to the East (West) of the ground-based station. Note that since similar results are found for day and nighttime measurements, only comparisons for day time data are shown in the following.

## 3.2 Comparison with FTIR and ozonesonde data

For the comparison between IASI data against FTIR and sonde TOCs and partial ozone columns, the coincidence criteria used in this study are the same as those defined in Boynard et al. (2016), except for the time coincidence which is slightly different in order to be more consistent with the temporal variability of tropospheric ozone: we apply coincidence criteria of 100 km search radius and ±6 h. As the ozonesonde measurements are mainly performed in the morning (local time), this implies that most of the pixels meeting these coincidence criteria correspond to pixels of the IASI morning overpass, which is not the case for FTIR measurements that can be performed all day long.

In the comparison with FTIR data, the FTIR retrieved profiles are adjusted following Rodger and Connor (2003, their Eq. 10) in order to take into account the different a priori profiles used in both IASI and FTIR retrievals:

$$x_{adjusted,FTIR} = x_{FTIR} + (\mathbf{A}_{FTIR} - \mathbf{I})(x_{a,FTIR} - x_{a,IASI}) \tag{1}$$

where $\mathbf{A}_{FTIR}$ is the FTIR AK matrix, $\mathbf{I}$ the unity matrix, and $x_{a,FTIR}$ and $x_{a,IASI}$ the FTIR and IASI $O_3$ *a priori* profiles, respectively.

In addition, when validating satellite profile products, a proper comparison method is to account for the difference in vertical resolution. In the present work, the ozonesonde and adjusted FTIR profiles are first interpolated on the corresponding IASI

vertical grid and then degraded to the IASI vertical resolution by applying the IASI AKs and *a priori* O$_3$ profile according to Rodgers (2000):

$$\boldsymbol{x}_s = \boldsymbol{x}_a + \mathbf{A}(\boldsymbol{x}_{raw} - \boldsymbol{x}_a) \tag{2}$$

where $\boldsymbol{x}_s$ is the smoothed ozonesonde/FTIR profile, $\boldsymbol{x}_{raw}$ is the ozonesonde/adjusted FTIR profile interpolated on the IASI
vertical grid (referred as "raw" FTIR), $\boldsymbol{x}_a$ is the IASI *a priori* profile and **A** the IASI AK matrix. Incomplete ozonesonde profiles above ozonesonde burst altitude are filled with the *a priori* profile.

For each ozonesonde/ FTIR measurement, we calculate the TOCs (only for the FTIR data) and the four partial columns defined above from all IASI and smoothed ozonesonde/FTIR profiles meeting the coincidence criteria, then we average all IASI and smoothed ozonesonde/FTIR total and partial columns. In the end there is one IASI-DATA profile pair per ozonesonde/FTIR
measurement. To avoid unrealistic statistics skewed by extremely unrealistic low values in the UTLS O$_3$ columns found in the smoothed ozonesonde data, we filter out extreme outliers exceeding 200 % relative differences with IASI (which can be up to ~8 % of the data in the tropical UTLS).

## 4  IASI-A and IASI-B O$_3$ consistency

Before validating IASI-A and IASI-B O$_3$ products, we assess the consistency between both instruments over the common
period May 2013 – July 2017. For the intercomparison exercise, we first calculate the daily IASI-A and IASI-B averages over a 1°x1° grid. Then for each 1°x1° grid cell, we calculate the relative difference as 100x[(IASI-A – IASI-B)/IASI-B]. Finally we calculate the monthly averaged data from the daily gridded differences. A statistical analysis of IASI-A and IASI-B TOCs and TROPO O$_3$ columns is performed with respect to time and latitude.

Figure 3 illustrates the 1° zonal monthly relative differences between IASI-A and IASI-B TOCs (computed from daily gridded
differences) for daytime measurements (left panel) and nighttime measurements (right panel). IASI pixels are considered as daytime or nighttime data if the solar zenith angle (SZA) is <90° or >=90°, respectively. An excellent agreement between both IASI-A and IASI-B TOCs is observed, with differences within 0.4 %, except for the polar regions. As already discussed in Boynard et al. (2016), a possible reason for the larger differences in polar regions is the combination of the overlap by consecutive orbits with different times and thus, different meteorological conditions. Metop, with its polar orbit, makes 14
revolutions per day, and will therefore pass by the poles on each revolution. This leads to a larger number of observations over the poles each day at different local times for the same grid cell. The variability in O$_3$ is therefore much larger leading to both larger differences between the measurements and larger standard deviation (not shown). Two interesting features that come out of Fig. 3 are (i) the slight increase in the differences in 2015 (April-September) and the decrease in the differences between the period prior to April 2015 and the period after September 2015. These two points will be discussed hereafter.
Figure 4 illustrates the 1° zonal monthly relative differences between IASI-A and IASI-B TROPO O$_3$ columns (computed from daily gridded data) for daytime measurements (left panel) and nighttime measurements (right panel). In general, the differences between IASI-A and IASI-B TROPO O$_3$ columns are within ±2 % although larger differences can be found locally,

especially in the polar regions. As for the TOCs product, the differences decrease from October 2015 with respect to the period May 2013 – April 2015 and the differences are significantly larger for the period April – September 2015 (up to 10 %). Another noticeable feature during the period April – September 2015 is the opposite signs between the differences in TOCs (Fig. 3) and in TROPO $O_3$ columns (Fig. 4).

The reason for these unexpected differences lies in the fact that on 13 April 2015, there was an error in the IASI-A pixel registration, which slightly modified the IASI-A viewing angle (Buffet et al., 2016). This was corrected only in September 2015 and produced a ~5-month period (between April and September 2015) with somewhat larger differences observed between IASI-A and IASI-B $O_3$ products. Furthermore, on 7 October 2015, the IASI's cube corner compensation device, which was shown to generate micro-vibrations and random errors in the IASI spectra, was stopped. As a result, since October

2015, the IASI-A and IASI-B spectra are of better quality/stability (Buffet et al., 2016; Jacquette et al., 2016).

Because of the changes made in the IASI-A Level-1 data processing, the comparison statistics are performed over two periods, excluding the period between April and September 2015: Over the period May 2013 – March 2015, IASI-A TOC product measures 0.3±1.1 % less ozone than IASI-B for both day- and nighttime measurements. From October 2015, as expected, the overall differences and standard deviation are smaller: IASI-A TOC product gives 0.1±0.5 % less ozone than IASI-B. Similar

results are found for the TROPO $O_3$ column: Before April 2015, IASI-A TROPO $O_3$ product gives 2.4±0.5 % and 2.1±0.4 % less than IASI-B for day- and nighttime measurements, respectively. From October 2015, the overall difference between both instruments decreases and is equal to 1.4±1.3 %.

The excellent agreement between the current IASI-A and IASI-B TOC and TROPO $O_3$ columns (April – September 2015 excluded) allows the combined use IASI-A and IASI-B instruments to provide homogeneous total and tropospheric ozone data

with full daily global coverage measurements. Even if for the period April – September 2015, IASI-B $O_3$ products are better recommended for a high quality use, it is worth noting that the IASI-A instrumental issue only affects the TOC by 0.4% and the tropospheric ozone by 10%, which are much lower than the TOC and tropospheric retrieval errors estimated to 2% and 15 % on average, respectively, justifying the potential use of the IASI-A data over that period if it is required. In the validation exercise presented in the next section, the period April-September 2015 is included.

The interannual variability of IASI-A TOCs and TROPO $O_3$ columns is illustrated in Fig. 5. Highest TOC occurs in the northern mid- and high latitudes during springtime while lowest TOC values (<200 DU) occur in the southern high latitudes from September to November. Lowest TROPO $O_3$ occurs southwards 70° S as well as in the tropics (values less than 15 DU), whereas monthly mean TROPO $O_3$ values occur in the northern mid-latitudes during summer, which is mainly caused by stratosphere-troposphere exchange process in spring-summer coupled with $O_3$ production from pollution events in summer.

## 5. Validation results

### 5.1 Comparison with GOME-2 TOCs

Figure 6 illustrates the 1° zonal monthly relative differences between IASI-A and GOME-2A TOCs (computed from daily data) for the period 2008 – 2017 with their associated standard deviation. A good agreement between both TOC products is observed, with the lowest differences found in the mid-latitudes and tropics and the largest differences found in the polar regions, especially over Antarctica (differences larger than 20 %). In the tropics the differences are mostly positive while they are negative in the mid-latitudes.

Figure 7 shows the seasonal distributions of relative differences between IASI-A and GOME-2A TOCs, computed from daily gridded data for the period 2008-2017 (see Table 2 for the associated statistics). The lowest differences are found in the northern mid-latitudes during summer (June-July-August) where the IASI sensitivity is the highest, while the largest differences are found over cold surface of Antarctica and Greenland where the IASI sensitivity is the lowest, especially during the March-April-May season (3.5 % over Antarctica). The detailed analysis undertaken for different latitude bands given in Table 2 shows that the highest correlation coefficients are found in the mid-latitudes and the northern high latitudes, with values higher than 0.93. Lower correlation is found between IASI-A and GOME-2A TOCs in the Southern high latitudes during MAM (0.62) and in the tropics during SON (0.55). However, during the $O_3$ hole season, high correlation of 0.94 is found in the southern polar region, with IASI-A TOCs being negatively biased (~2%). This suggests that IASI-A TOC overestimates the extent of $O_3$ depletion (i.e. underestimates the TOCs in the ozone hole) with respect to GOME-2A TOC.

Figure 8 illustrates the time series of the monthly mean relative difference between IASI-A and IASI-B against GOME-2A TOCs along with the standard deviation for the Northern Hemisphere (NH) and the Southern Hemisphere (SH).There is a pronounced seasonality in the difference between IASI-A and IASI-B against GOME-2A TOCs in the SH, with the largest differences being found during austral summer (up to 4 %) and the lowest differences during the austral winter. Compared to GOME-2A data, IASI-A (IASI-B) TOC shows less $O_3$ in the NH by 0.20±0.74% (0.15±0.69%) and more $O_3$ in the SH by 0.42±1.42% (0.28±1.87%), these differences being in the total retrieval error bars of the two products. Globally, IASI-A (IASI-B) TOC product are slightly higher than GOME-2A TOC product, with a global mean bias of 0.3±0.8 % (0.4±0.8 %). It is worth noting that the previous IASI TOC product (v20140922) was in disagreement by more than 5 % (Boynard et al., 2016). The global mean bias is now within total errors of GOME-2 estimated to 3-7 % (Valks et al., 2017) and IASI, which demonstrates the good consistency between IASI and GOME-2 TOC products.

Despite the global improvement of ~5 % with the new IASI TOC product with respect to the previous IASI TOC product (v20140922), large discrepancies are still observed at high latitudes and are partly explained by:

i)      the low spectral signal to noise ratio due to very low surface temperature in this region leading to limited information content in the IASI observations in these regions;

ii) a misrepresentation of the wavenumber-dependent surface emissivity, which is a critical input parameter to describe the surface, especially above continental surfaces (Hurtmans et al., 2012). FORLI uses the emissivity climatology built by Zhou et al. (2011) providing weekly emissivity values on a 0.5°x0.5° latitude/longitude grid for all 8461 IASI spectral channels. However, Zhou et al. climatology can have missing values. In such cases, the MODIS climatology built by Wan (2006), which provides values for only 12 channels in the IASI spectral range is used instead. Furthermore, in case of no correspondence between the IASI pixel and either climatologies, the reference emissivity used for the Zhou climatology (Zhou et al., 2011) is used, which can significantly impact the retrievals, in particular in arid or semi-arid regions where variations in emissivity are large both on spectral and spatial scales (Capelle et al., 2012) but also in ice region since the reference emissivity does not necessarily reflect the actual snow or sea ice coverage;

iii) the temperature profiles used in FORLI-$O_3$ that are less reliable at high latitudes and over elevated terrain (August et al., 2012). As shown in Boynard et al. (2009), the errors introduced by the uncertainties of 2 K on the temperature profile can reach up to 10 % of total error on the retrieved vertical profile, with the error due to the temperature uncertainty on the TOCs being much lower. Errors on thermal contrast can also have an impact on the retrievals.

iv) the errors associated with TOC retrievals in the UV-vis spectral range increasing at high solar zenith angles in these regions, mostly because of the larger sensitivity of the retrieval to the *a priori* $O_3$ profile shape (Lerot et al., 2014).

In the section below, a detailed analysis of the larger bias found in the Antarctic region is undertaken for individual ground-based Brewer and Dobson station to try to understand the larger bias (see next section).

Because of GOME-2 instrumental degradation for several years (Dikty and Richter, 2011), the stability of IASI-A and -B is not assessed from comparison with GOME-2A. It will be explored in subsections below against the other independent datasets used in this study.

## 5.2 Comparison with Brewer/Dobson TOCs

Figure 9 shows the dependency of the relative differences of IASI-A and IASI-B against GB measurements on latitude, for the period May 2013 – July 2017. For each daily ground-based measurement a relative difference is calculated as 100 x (IASI – GB) / GB [%]. All relative differences are then separated into latitudinal bins of 10° and the mean is calculated. As expected, very similar features between the IASI-A and IASI-B comparisons can be seen, with the Antarctic (80° S-90° S latitude band) being largely overestimated (~20 %) and the northern middle latitudes driving the mean comparisons around the 0 % to 2 % level. As shown by the IASI-to-Dobson comparison (left panel), the dependency on latitude is less visible for the NH due to the high number of collocations which renders the latitudinal means more representative compared to the SH. The comparisons with Dobson measurements show differences between 0 and 2.5 % for the entire NH (except in the 70-80°N belt where

difference reaches 3.5 % for IASI-A) and for latitudes ranging between 0° and 40° S. Southwards of 40°S, the differences range between 2 and 4 %, which is partially attributed to the small number of stations, the limited sensitivity in this region (especially for latitudes lower than 60°S) and the larger TOC variability within the Southern polar vortex (Garane et al., 2018, this issue; Verhoelst et al., 2015). A similar picture for the NH is observed for the comparison with Brewer measurements.

Note that there are a few Brewer stations in the SH, but they are not evenly distributed (all of them are located on the Antarctic) so their measurements are not used. From Figure 9 we can also notice the larger differences for the 20-30°N latitude band (more visible for the comparison with Brewer measurements), where some desert stations, like Tamanrasset, Algeria and Aswan, Egypt (see further discussion in the next paragraph) are located, which suggests that the IASI quality flag established to filter the high values linked with emissivity-related issues (based on the ratio of the surface-6 km column relative to the

TOC) is rather loose. Nevertheless the overall comparison with Dobson and Brewer TOCs shows that IASI new TOC product is improved by 4 % in comparison with the previous IASI TOC product (v20140922; see Boynard et al. (2016)) and is within IASI and GB TOC total error bars.

To further examine the large discrepancies mentioned above, we have analyzed in more details the results obtained for individual stations located in Antarctic and desert regions. The stations located near desert areas show an diverging behavior

with positive (Tamanrasset, Algeria) and negative (Aswan, Egypt and Springbok, South Africa) biases of +7 to +8 % and -5 to -4 %, respectively. Over Antarctica, four stations were examined: the bias was found to be extremely high for Amundsen-Scott located at 90° S and 3 km altitude (~20 %) and less, but still positive, for the other three stations Haley-Bay, Syowa, Arrival-Heights (1.2–3.8 %) located on the Antarctic Ice Sheet. The comparison of GOME-2A with ground-based TOCs at Amundsen-Scott shows a very small bias of 1-2 %, indicating there is no obvious issue with the ground-based measurements.

Furthermore, the scatter plot for that particular station (compared to either Dobson or Brewer; plot not shown) shows that IASI-A has a much higher variability than the GB TOC values. This issue has still to be further explored by investigating, for instance, the impact of potential surface emissivity discrepancies on the retrievals over some regions of Antarctica and deserts. Additional quality filters, e.g. on ice surface emissivity issues, could also be considered.

Figure 10 shows the time series of the monthly relative differences between IASI-A, -B and GB TOC over the corresponding

IASI measurement period for the NH only. For each GB measurement, a daily relative difference is calculated. All the relative differences are then averaged per month. Each month includes more than 180 IASI-GB pairs. As for GOME-2, we can see an obvious seasonal variability in the differences, especially for the Dobson measurements: the smallest differences appear in summer and the largest differences in winter. The larger seasonal variability in the Dobson comparisons is explained by the fact the Dobson measurements strongly depend on the stratospheric effective temperature (Koukouli et al., 2016). We can also

see a similar but less pronounced seasonality effect in the Brewer comparison. According to Garane et al. (2018, this issue) and references therein, even though Dobson and Brewer spectrometers follow almost the same principles of operation, TOC measurements from the two types of instruments show differences in the range of ±0.6 % due to the use of different wavelengths in their respective TOC algorithms and the different temperature dependence for the ozone absorption coefficients. However it is worth noting that these differences between Brewer and Dobson TOCs are lower than their total

uncertainty (~1 %). The mean difference for the NH is lower than 1.1 % for both Dobson and Brewer comparisons to the IASI observations.

According to the user requirements given in the User Requirement Document of the Ozone_cci project (van Weele et al., 2016), the stability of the ozone measurements must be among 1 and 3 % decade$^{-1}$. To assess the long-term stability of the
IASI-A TOC products, which is essential for trend studies, we calculate the IASI-A TOC decadal drift from the monthly relative differences between IASI-A and GB TOC over the period 2008 – 2017 (see Fig. 10). The drift is considered statistically significant if its $P$ value is lower than 0.05 and the drift value is higher than its 2-σ standard deviation. For the Dobson comparison, the TOC relative differences exhibit insignificant drift of 0.68±0.69 % decade$^{-1}$. For the Brewer comparison, a <3 % positive drift of 1.38±0.50 % decade$^{-1}$ is found. When comparing against Brewer and Dobson measurements, the results
show that the IASI-A TOC products are stable and thus reliable for trend studies, as expected from the excellent stability in the Level-1 (Buffet et al., 2016).

## 5.3 Comparison with SAOZ TOCs

Figure 11 shows the temporal variation of the day time monthly mean relative differences between IASI-A and IASI-B against SAOZ TOCs for the eight SAOZ stations for the period 2008 – 2017. For each daily SAOZ measurement, a relative difference
is calculated as 100 x (IASI – SAOZ) / SAOZ [%]. All the relative differences are then monthly averaged. First, we clearly see the systematic seasonality in the differences, with increasing amplitude with latitude. Compared to SAOZ, the IASI-A and IASI-B TOCs are biased by 0.5-2 % (~1 % monthly mean averaged standard deviation) in the tropics and mid-latitudes, and biased high to about 4±3 % inside the polar circle. The results are consistent with those found for the comparison with GOME-2A along with Brewer and Dobson measurements (see Sections 5.1 and 5.2, respectively). An improvement of 3-4 % is found
when compared to the previous IASI product (v20140922).

The IASI-A and SAOZ TOC relative differences show small or insignificant negative decadal trends ranging between -0.05±0.70 % (OHP) and -2.27±0.71 % (Reunion), except for Bauru station, which is due to SAOZ retrieval issue still under investigation. The good quality of the IASI-A TOC temporal stability satisfies well the 1 – 3 % decade$^{-1}$ Ozone_cci requirements for the long-term stability for total ozone measurements (Van Weele et al., 2016), which shows again that the
current IASI-A TOC products are homogeneous and reliable for trend studies.

## 5.4 Comparison with FTIR TOCs and partial ozone columns

Figure 12 shows the temporal variation of the monthly mean relative differences between IASI-A and IASI-B against FTIR TOCs convolved with the IASI averaging kernels according to Eq. (2) for the six FTIR stations (see Table 1 and Fig. 2 for their location) for the period 2008 – 2017. Compared to FTIR, the IASI-A and IASI-B TOCs are negatively biased by 0.8-
6.2 %  with the largest biases (-4.1 % and  -6.2 %) at Jungfraujoch and Lauder, respectively. At Lauder, mean biases of 5.7±5.4 % and 0.6±6.4 % between FTIR and IASI against Dobson TOCs, respectively, are found, suggesting that the FTIR

data might be biased high at that station, but 4 % of this bias between FTIR and Dobson is likely due to the known inconsistency between IR and UV cross-sections (Gratien et al., 2010) (note that the bias is calculated as [100x(FTIR-DOBSON)/DOBSON] or [100x(IASI-DOBSON)/DOBSON]). It can be noted that the bias between FTIR and IASI-A, and SAOZ and IASI-A for close latitude stations are very consistent, if one takes this spectroscopic bias into account (i.e. UV Sodankyla lower than IASI-A by 3.9%, FTIR Kiruna higher by 1.1 %; UV OHP lower than IASI-A by 1.0 %, FTIR Jungfraujoch higher by 3 %; UV Kerguelen higher than IASI-A by 0.9 %, FTIR Lauder higher by 6.2 %).

At Zugspitze and more particularly at Jungfraujoch, two jumps are visible in 2010 and 2014, with larger biases before 2011 and after 2014 with respect to the period in between. It is worth noting that these two jumps seem to coincide with changes in IASI L2 temperature (in September 2010 and September 2014). The analysis of surface temperatures used in both IASI (Eumetsat) and FTIR (NCEP) retrievals (IASI L2 Eumetsat and NCEP, respectively) shows that the differences between Eumetsat and NCEP can reach up to 20 K for the surface temperature and vary between -10 and 10 K along the temperature vertical profile at both Jungfraujoch and Zugspitze while at the other stations the differences are much lower (less than $|5|$ K), which suggests that IASI L2 Eumetsat temperatures are less reliable above elevated areas. However a more in-depth analysis is needed and for that matter is in progress in order to understand the exact origin of the jumps found in the differences between IASI and FTIR TOCs at these stations.

The dominant systematic uncertainty in FTIR $O_3$ retrievals is due to the spectroscopic parameters (García et al., 2012). The IASI retrieval algorithm uses HITRAN 2012 and the FTIR retrieval algorithm uses HITRAN 2008, however no differences were found in the $O_3$ absorption band, respectively (Boynard et al., 2016). We do not expect a significant bias between the IASI and FTIR total columns due to ozone spectroscopy, because both retrieval algorithms use the same ozone spectroscopic parameters and the same fitting spectral range. Except at Lauder and Jungfraujoch, the mean biases between IASI and FTIR TOCs are relatively low and within total errors of FTIR (e.g. García et al., 2012) and IASI, which shows again the good quality of IASI TOC data.

Except at Jungfraujoch and Zugspitze, the IASI-A and FTIR TOC monthly relative differences show insignificant drift less than $|0.9|$ % decade$^{-1}$ (see Fig.12 and Table 2), which is among the $1 - 3$ % decade$^{-1}$ Ozone_cci requirements for the long-term stability for total ozone measurements (Van Weele et al., 2016), demonstrating that the current IASI-A TOC products are homogeneous and reliable for trend studies. The significant negative drifts found at Jungfraujoch and Zugspitze, are explained by the bias drop observed from 2014 that is discussed above.

Since FTIR data also provide up to four independent pieces of information in the vertical ozone profile, we now assess four IASI partial ozone columns characterized by a DOFS of ~1 (surface-300 hPa, 300-150 hPa, 150-25 hPa and 25-3 hPa), which should make such assessment meaningful. The comparisons of the four partial ozone columns between IASI-A and FTIR performed for the period 2008 – 2017 are presented in Fig. 13. The correlation coefficients between FTIR and IASI-A partial columns are good to excellent (from 0.72 to 0.98), with the highest correlations found in the UTLS and LMS.

For all stations except Kiruna, IASI tropospheric column is negatively biased by 5-14 %. The comparison for the UTLS $O_3$ columns shows that IASI-A $O_3$ product is positively biased at all stations (except at Izaña), with the largest bias found at

Wollongong (21.1±19.9 %) and the lowest bias found at Jungfraujoch (3.7±15.0 %). The standard deviation is maximum in the UTLS at Izaña and Lauder, which is due to strong $O_3$ variability and large total retrieval error in this region as shown in Wespes et al. (2016). Indeed their Fig. 4b demonstrated that in tropical regions the estimated total retrieval error of vertical ozone profiles from IASI are larger than in middle latitudes, which suggests that it would be the case for the ozone column as

well. It should be noted that IASI is positively biased in the UTLS region, as reported in previous studies comparing IASI to ozonesonde data (e.g. Boynard et al., 2016; Dufour et al., 2012; Gazeaux et al., 2013). Although Dufour et al. (2012) attempted to give some explanations for this particular feature, the exact reason for this overestimation is still not clear. One reason could be the use of inadequate *a priori* information. Note that FORLI uses only one single *a priori* profile (Hurtmans et al., 2012) that is the global mean profile of the McPeters/Labow/Logan climatology (McPeters et al., 2007). As shown by Bak et al.

(2013), using tropopause-based ozone profile climatology can significantly improve the a priori. However, using dynamical *a priori* makes the comparison on a global scale less straightforward since a different a priori profile would be used at each IASI pixel. The best correlation coefficients and smaller standard deviations (in %) between IASI-A and FTIR data are found for the LMS column. The small standard deviations in the LMS comparisons allow the detection of consistent IASI-A negative biases at all stations (5-9%). This consistent negative bias in the LMS, where the ozone partial column contributes the most to

the total column, is reflected in the observed negative bias on TOC discussed above. These better correlation coefficients and standard deviations in LMS are due to the better IASI sensitivity to this column (mean DOFS ~1.2 – 1.5 as indicated in Fig. 13) compared to the other partial columns. The smallest biases between FTIR and IASI-A columns are found in the MS column (-0.2 / +4.9%), except at Kiruna where the bias reaches 13 %. This higher bias at Kiruna might be due to a bad collocation of sounded air masses which can be in different in or out polar vortex conditions for the two instruments. The FTIR instrument

sounds the atmosphere along the line-of sight instrument-sun, therefore the sounded air masses at this higher partial column and for high solar zenith angles measurements might be far away from the station itself (few hundreds kilometers). A collocation with the satellite that would take the FTIR line-of sight into account, would improve the comparisons.

A similar picture us found for the comparison between IASI-B against FTIR partial ozone columns over the period May 2013-2017 (not shown).

The stability of IASI-A partial ozone columns is also assessed based on the time series of monthly relative differences between IASI-A and FTIR data over the period 2008 – July 2017. Table 3 gives the decadal drift values along with their 2-σ standard deviations in % decade[-1] as well as the P-value. As a reminder the trend is considered significant if the drift value is higher than its 2-σ standard deviation. For the TROPO column, we clearly see a significant negative drift at all stations ranging from -5.0±4.8 % decade[-1] (Izaña) to -16.1±8.1 % decade[-1] (Kiruna). Smaller or insignificant drifts are found in the UTLS and LMS.

Regarding the MS, insignificant positive drifts are found, except at Izaña where a positive drift is found (3.7±2.5 % decade[-1]). As a consequence, the stability of the IASI-A partial $O_3$ columns when compared to the six FTIR GB measurements that cover the IASI measurement period and that are characterized by limited vertical sensitivity cannot be confirmed.

To answer that question, comparisons of IASI partial $O_3$ columns with ozonesonde measurements that provide numerous highly resolved vertical $O_3$ profiles is performed in the section below.

## 5.5 Comparison with ozonesonde partial ozone columns

A statistical comparison of IASI-A and IASI-B against sonde partial ozone columns at 56 stations (see Fig. 2) is performed, which gathers approximatively 2000 ozonesonde profiles during a period extending from May 2013 to July 2017 and 11600 ozonesonde profiles over the whole IASI measurement period (2008 – 2017). In order to assess the latitudinal variability of IASI $O_3$ retrieval performance, the comparison is performed for six 30° latitude bands representative of the northern high latitudes (60-90°N), northern mid-latitudes (30-60° N), northern tropics (0-30°N), southern tropics (0-30°S), southern mid-latitudes (30-60°S) and southern high latitudes (60-90°S).

Figure 14 shows the comparison of IASI-A against smoothed ozonesonde for four partial columns for each of the six-latitude bands during 2008 – 2017. For the TROPO $O_3$ columns (1st column), the mean biases and standard deviation are within 20 %, IASI-A underestimating the $O_3$ abundance in the tropics and mid-latitudes (by ~16-19 % and ~6-11 %, respectively) and overestimating the $O_3$ abundance at high latitudes (by 4-5 %), compared to ozonesonde data. The correlation coefficient ranges from 0.8-0.9 in the tropics to 0.7-0.8 at middle latitudes, and from 0.5 to 0.8 at high latitudes. The linear regression slopes are in the range $0.6 – 0.8$, with lower values found at high latitudes due to the reduced retrieval sensitivity to the lower troposphere. It is worth noting that a lower correlation coefficient is found for the southern mid-latitudes, which is likely due to the lower amount of data in comparison with the other latitude bands. The comparison for the UTLS $O_3$ columns (2nd column) shows that IASI-A $O_3$ products overestimate the $O_3$ abundance irrespective the latitudes, with the largest biases found in the high latitudes (30-42 %) and the lowest biases found in mid-latitudes (~11-19 %). The standard deviation is maximum in the UTLS in all latitude bands (compared to the other partial columns) due to strong $O_3$ variability and large total retrieval error as shown in Wespes et al. (2016).The linear regression slopes are close to 1 in the polar and mid-latitude regions but are around 0.4 in the tropics, which is closely related to the small amount of $O_3$ in the tropical UTLS. A positive bias from IASI-A $O_3$ products is also found for the LMS (3rd column) and MS (4th column) columns (except for the high latitudes for the latter). The correlation coefficient ranges between 0.6 (tropics and high latitudes) and 0.8 (mid-latitudes) for the LMS column while they are much lower for the MS column, which is explained by the low DOFS values ranging between 0.4 and 0.6 as indicated on the scatter plots. Note that the DOFS for the MS columns are lower than those calculated in Fig. 13 because they do not correspond to the full MS column calculated from IASI (25-3 hPa i.e. ~25-40km) but to the MS columns truncated to match the maximum altitude (30-35 km) of the sonde measurements. The mean DOFS is generally in the range $0.6 – 1.4$ for the TROPO, UTLS and LMS columns, the larger DOFS being found for the LMS column. Similar results are found for the comparison between IASI-A and IASI-B against sonde partial ozone columns over the common period May 2013- 2017, except for the MS in the 60-90°S latitude band (not shown). In comparison with the previous IASI partial ozone column products reported in Boynard et al. (2016), the new IASI ozone product is significantly improved in the MS by 8-12 % for the mid latitudes and tropics. The improvement is less significant for the LMS except in Antarctic where an improvement of 6 % is found. As for the TROPO and UTLS columns, no or slight improvement (<2 %)  is found, and the agreement between IASI

and sonde data is even worse compared to the previous IASI ozone product, especially for the southern tropical TROPO column (by 7 %) and the UTLS column (by 10-18 %).

Figure 15 illustrates a sample of time series of daily IASI-A and smoothed ozonesondes TROPO $O_3$ columns along with the corresponding differences for six ozonesonde stations representative of different latitude bands over the period 2008 – 2017. The comparison is good for all latitudes, with IASI-A $O_3$ products underestimating the TROPO $O_3$ abundance in the mid-latitudes and tropics by ~1.7-3.5 DU (5.5-10.1 %) and overestimating the TROPO $O_3$ abundance in the high latitudes by ~1.5 DU (5–7 %). This result is generalized in Fig. 2, which shows the mean and standard deviation of the differences in DU of TROPO $O_3$ columns between IASI-A and smoothed ozonesonde for each ozonesonde station used in the present work over the period 2008 – 2017. Overall, IASI-A TROPO $O_3$ product exhibits good agreement with ozonesonde data at most of the stations, with mean relative difference and standard deviation within |6| DU. An interesting feature seen in Fig. 2 is that the mean and standard deviation of the differences of TROPO $O_3$ columns between IASI-A and smoothed FTIR is lower than those between IASI-A and sonde TROPO $O_3$ column.

The long-term stability of IASI-A partial $O_3$ column vs ozonesonde measurements is assessed in Figure 16, which presents the monthly relative differences between IASI-A and ozonesonde for the TROPO, UTLS, LMS and MS $O_3$ partial columns for a total of 18 ozonesonde stations in the NH that cover eight years or longer (over 2008 – 2017). With more than 30 IASI-sonde pairs per month, the NH presents sufficient collocated data to assess a good statistical drift analysis on the contrary to the SH (only 8 ozonesonde stations). For each ozonesonde measurement, a daily relative difference is calculated. All the relative differences are then monthly averaged. A main feature that arises from this figure is the pronounced seasonality in the differences between IASI-A and sonde $O_3$ for the UTLS and LMS column, with the lowest differences found in summer and the largest differences found in winter. We can also see a small but apparent seasonality in the differences for the TROPO $O_3$ column: the IASI TROPO $O_3$ column appears less biased with respect to the ozonesondes during winter. This reflects the low sensitivity of IASI associated with low brightness temperature in the troposphere and in such situations, the IASI retrieval mostly provides the *a priori* information (see Eq. 2). The differences in the TROPO $O_3$ column are better than -10 % during the period 2008 – 2010 and decrease up to -20 % from 2011. This feature is also visible for the MS column: the difference baseline is around the 0 % level between 2008 and 2010 but near the 4 % level from 2011.

The linear trends of the monthly mean ozone biases for each partial column are plotted in Fig. 16 for the period 2008 – 2016 (blue line). Note that 2017 is not included in the drift calculation because of lower number of collocated data for that year. Based on the drift value with the 2-$\sigma$ standard deviation and the $P$ value (indicated on each plot), the derived trends are insignificant for the UTLS and LMS but are statistically significant for the TROPO and MS columns (-8.6±3.4% decade$^{-1}$ and ~5.4±3.6% decade$^{-1}$, respectively), which is in agreement with Keppens et al. (2018, this issue) who applied a different method based on bootstrapping technique (Hubert et al., 2016). Note that for the TROPO column, the drift calculated for each individual station ranges between -16 % decade$^{-1}$ and -5 % decade$^{-1}$, which is the same order of magnitude of those found in the IASI-A-to-FTIR TROPO comparison. If we limit the time period to 2011 – 2016, no statistically significant drift is found

anymore for the TROPO and MS (P value >0.47), as expected from the excellent stability in the Level-1 (Buffet et al., 2016). However, since this difference in the drift values might be due only to the too short time periods considered here associated with the high variability in the TROPO $O_3$ differences, a few more years are needed to confirm the observed negative drifts and evaluate them on the longer term.

## 6 Summary

In this study, we have assessed the quality of IASI-A and IASI-B $O_3$ products (total and partial columns) retrieved with the FORLI v20151001 software for nine years (2008 – 2017) through an extensive inter-comparison and validation exercise using independent observations (satellite, ground-based and ozonesonde). Compared to the previous version of FORLI-O3 (v20140922), several improvements were introduced in FORLI-O3 v20151001, including absorbance look-up tables recalculated to cover a larger spectral range using the 2012 HITRAN spectroscopic database (Rothman et al., 2013), with additional numerical corrections. This leads to a change of ~4% in the Total Ozone Column (TOC) product, which is mainly associated with a decrease in the retrieved $O_3$ concentration in the middle stratosphere (above 30 hPa/25 km). The IASI $O_3$ products processed with FORLI v20151001 are part of the ESA Ozone_cci and ECMWF C3S projects, which focus on building consolidated climate-relevant ozone data sets as ECVs. Therefore, validating the latest version of the IASI $O_3$ products over a long-time period and assessing their stability are necessary for decadal trend studies, model simulation evaluation and data assimilation applications. The main findings of this work can be summarized as follow:

1. The inter-comparison between IASI-A and IASI-B TOC products for the period May 2013 – July 2017 shows that, IASI-A and IASI-B TOCs are consistent, with a global difference less than 0.3 % for both day- and nighttime measurements and with IASI-A TOCs slightly higher than those of IASI-B. A similar result is found for the TROPO $O_3$ column: a global difference less than 2.4 % for both day- and nighttime measurements is found, IASI-A TROPO $O_3$ columns lower than IASI-B. Inconsistencies between both instruments were found for a limited period between April and September 2015, which are due to the change in the IASI-A viewing angle that was corrected in September 2015 (Buffet et al., 2016). However, it is worth noting that the impact of IASI-A instrumental issue is within the TOC and TROPO $O_3$ column retrieval error bars. In case of using IASI-A data only, the user is free to include or exclude the period April – October 2015 depending on the interest of the study. The consistency between IASI-A and IASI-B $O_3$ products becomes better after September 2015 (differences less than 0.1 % and 1.4 % for the TOC and TROPO $O_3$ column product, respectively), which is due to the better quality of IASI-A and IASI-B Level-1 data because of the stop of IASI's cube corner compensation device, which proved to generate micro-vibrations and random errors (Buffet et al., 2016; Jacquette et al., 2016).

2. With respect to GOME-2A data, IASI-A and IASI-B TOCs are in excellent agreement: they are marginally lower in the Northern Hemisphere by 0.2 % while they are higher in the Southern Hemisphere by 0.4 %. There is a pronounced seasonality in the differences in the SH, with the largest differences found during the austral summer (up to 4 %) and

related to larger differences at the southern high latitudes. With respect to Dobson and Brewer data, IASI-A and IASI-B TOC product overestimates the total $O_3$ abundance by 0.5-1.1 % with an obvious seasonal variability in the differences, which is caused by the ground-based measurements (see Section 5.2 for more explanation). Compared to SAOZ, IASI-A and IASI-B TOC product is biased by 0.6-2 % (~1 % monthly mean averaged standard deviation) in the tropics and mid-latitudes, and this value is increasing to about 2.5-3.8 % inside the polar circles. Finally, a good agreement is found between IASI-A and IASI-B against FTIR TOC product, with IASI underestimating the TOC by 1.1–6.2 %, the largest bias being found at Lauder, which is likely due to FTIR data that might be biased high by 1.5-2% at that station. It can be noted that the bias between FTIR and IASI-A, and SAOZ and IASI-A for close latitude stations are very consistent, if one takes this spectroscopic bias into account (i.e. UV Sodankyla lower than IASI-A by 3.8 %, FTIR Kiruna higher by 1.1%;  UV OHP lower than IASI-A by 0.9 %, FTIR Jungfraujoch higher by 3.0 %; UV Kerguelen higher than IASI-A by 0.7%, FTIR Lauder higher by 5.6 %).

3.  The time series of relative differences between IASI-A against UV-vis GB TOCs show insignificant negative drift in the NH ($0.68\pm0.69$ % decade$^{-1}$ and P-value= 0.05) and small negative trend in the SH ($1.48\pm0.53$% decade$^{-1}$ and P-value=0.00), which satisfies the $1 - 3$ % decade$^{-1}$ Ozone_cci requirements for stability of ozone measurements. Similar results are found with the IASI-A/FTIR TOC comparison. This demonstrates the long-term stability of the current IASI-A TOC products.

4.  The comparison results between IASI-A and IASI-B against smoothed FTIR and ozonesonde partial $O_3$ columns vary in altitude, with maximum standard deviation for the UTLS (20-40 %) due to strong ozone variability and larger total retrieval errors (Wespes et al., 2016). Attempt of explanations for the larger bias found in the UTLS are given in Dufour et al. (2012) but no clear reason was found. A possible explanation could be the use of inadequate *a priori* information in that layer. The current version of FORLI uses as *a priori* profile a single global profile that is the mean of the McPeters/Labow/Logan climatology (McPeters et al., 2007). As shown by Bak et al. (2013), using tropopause-based ozone profile climatology can significantly improve the *a priori*. However, using dynamical a priori makes the comparison on a global scale less straightforward to analyze because the retrieval at each IASI pixel would be based on different a priori profiles. The IASI-A and IASI-B TROPO $O_3$ products underestimate the $O_3$ abundance in the mid-latitudes and the tropics (by 11-13 % and 16-19 %, respectively) and overestimates the $O_3$ abundance in the high latitudes (by 4-5 %).

5.   The IASI-A-to-FTIR TROPO $O_3$ column comparison exhibits significant negative trends ranging between -8 and -16 % decade$^{-1}$ over the period 2008 – 2017 at all stations. A significant negative trend of $-8.6\pm3.4$% decade$^{-1}$ is also found in the IASI-A to ozonesonde TROPO $O_3$ column comparison for the Northern Hemisphere. The observed negative drifts in the IASI-A TROPO columns might partly explain the apparent disagreement between the ozone tropospheric trends observed by IASI and GOME/OMI in the TOAR report (Gaudel et al., 2018). However, further investigation should be done since the TROPO columns are not calculated in the same way in the two studies. When considering the period 2011 – 2016, the drift value for the TROPO column decrease and become statistically

insignificant. However, since this difference in the drift values might be due only to the too short time periods considered here associated with the high variability in the TROPO $O_3$ differences, a few more years are needed to confirm the observed negative drifts and evaluate them on the longer term. However, the observed negative drifts of IASI-A TROPO $O_3$ product (8-16% decade$^{-1}$) over 2008 –2017 might be taken into consideration when deriving trends from this product and this time period.

6. The IASI-A TOC relative differences against independent measurements showed small or insignificant negative decadal drifts for the period 2008-2017, which indicates that the current IASI-A TOC products are homogeneous and reliable for trend studies. The IASI-A TROPO $O_3$ relative differences against sonde and FTIR data showed significant negative drifts for the period 2008-2017. It is therefore recommended for trend studies to wait for the new homogeneous IASI climate time series, which will be reprocessed using the ECMWF ERA5 temperatures reanalysis (Hersbach and Dee, 2016) and the reprocessed IASI Level-1 data.

The IASI-A and IASI-B $O_3$ products (total and vertical profiles) starting in October 2007 are generated by the LATMOS and ULB in a near-real time mode using FORLI-$O_3$ v20151001. Both IASI-A and IASI-B $O_3$ products retrieved using FORLI-$O_3$ v20151001 are already part of the Eumetsat's AC SAF Official Validation Monitoring found in lap3.physics.auth.gr/eumetsat/ as part of the operational Eumetsat services. This $O_3$ retrieval algorithm (FORLI-$O_3$ v20151001) is currently being implemented into the Eumetsat processing facility under the auspices of the AC SAF project in order to operationally distribute Level-2 IASI $O_3$ data to users through the EumetCast system in 2018.

## 7 Data availability

The IASI $O_3$ data processed with FORLI-$O_3$ v20151001 can be downloaded from the Aeris portal (http://iasi.aeris-data.fr/O3/; Aeris, 2017). The GOME-2 $O_3$ data are available on the AC SAF website (http://acsaf.org; AC SAF, 2017). The ozonesonde data can be downloaded from the WOUDC database (https://doi.org/10.14287/10000008; WMO/GAW Ozone Monitoring Community, 2017a) and from the NOAA-ESRL database (http://www.esrl.noaa.gov/gmd/dv/ftpdata.html; NOAA, 2017). The Brewer and Dobson soundings can be downloaded from the WOUDC database (https://doi.org/10.14287/10000004; WMO/GAW Ozone Monitoring Community, 2017b). The SAOZ data are available at http://saoz.obs.uvsq.fr (SAOZ, 2017). The FTIR data are available at http://www.ndacc.org (FTIR, 2018).

**Acknowledgments**

IASI is a joint mission of Eumetsat and the Centre National d'Etudes Spatiales (CNES, France). The IASI Level-1C data are distributed in near real time by Eumetsat through the EumetCast system distribution. The authors acknowledge the Aeris data infrastructure (https://www.aeris-data.fr/) for providing access to the IASI Level-1C data and Level-2 temperature data used in this study. This work was undertaken in the framework of the Eumetsat AC SAF project (http://acsaf.org/), the European Space Agency $O_3$ Climate Change Initiative (Ozone_cci, www.esa-ozone-cci.org) and the Copernicus Climate Change (C3S). The French scientists are grateful to CNES and Centre National de la Recherche Scientifique (CNRS) for financial support. The research in Belgium is also funded by the Belgian State Federal Office for Scientific, Technical and Cultural Affairs and the European Space Agency (ESA Prodex IASI Flow project). The ozonesonde data used in this study were provided by the World Ozone and Ultraviolet Data Centre (WOUDC) and the Global Monitoring Division (GMD) of NOAA's Earth System Research Laboratory and are publicly available (see http://www.woudc.org and http://www.esrl.noaa.gov/gmd). The authors thank all those responsible for the WOUDC and GMD measurements and archives for making the ozonesonde data available. The ground-based total ozone column data used in this publication were obtained as part of WMO's Global Atmosphere Watch (GAW) and are publicly available via the WOUDC. We would like to acknowledge and warmly thank all the investigators that provide data to these repositories on a timely basis, as well as the handlers of these databases for their upkeep and quality guaranteed efforts. The FTIR data used in this publication were obtained as part of the Network for the Detection of Atmospheric Composition Change (NDACC) and are publicly available (see http://www.ndacc.org). The multi-decadal monitoring program of ULiege at the Jungfraujoch station has been primarily supported by the F.R.S.-FNRS and BELSPO (both in Brussels, Belgium) and by the GAW-CH programme of MeteoSwiss. The International Foundation High Altitude Research Stations Jungfraujoch and Gornergrat (HFSJG, Bern) supported the facilities needed to perform the FTIR observations. NDACC data from Kiruna were recorded at the Swedish Institute of Space Physics (IRF) Kiruna. The authors would like to thank Uwe Raffalski and Peter Völger from IRF Kiruna and Thomas Blumenstock, Frank Hase and Jochen Gross from KIT Karlsruhe for performing and analysing the FTIR measurements at Kiruna. NDACC data from Izaña were recorded at the Izaña Observatory on Tenerife Island. The authors would like to thank Omaira Garcia and Eliezer Sepúlveda, AEMet Spain, and Matthias Schneider, Frank Hase and Thomas Blumenstock from KIT Karlsruhe for performing and analysing the FTIR measurements at Izaña. The authors acknowledge the European Communities, the Région Réunion, CNRS, and Université de la Réunion for their support and contribution in the construction phase of the research infrastructure OPAR (Observatoire de la Physique de l'Atmosphère à la Réunion). OPAR is presently funded by CNRS (INSU) and Université de la Réunion, and managed by OSU-R (Observation des sciences de l'Univers à la Réunion, UMS 3365). Françoise Posny, Thierry Portafaix (LACy – UMR 8105) and Jean-Marc Metzger (UMS 3365) are also acknowledged for their management, scientific follow up and technical handling of the SAOZ and ECC observations at La Réunion.

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

**Tables**

**Table 1. List of the FTIR stations used for the validation of IASI TOCs and partial ozone columns. The latitude, longitude and altitude above sea level in kilometers (km a. s. l.) are provided for each station.**

| Stations | Latitude | Longitude | Altitude (km a. s. l.) |
|---|---|---|---|
| Kiruna | 67.8° N | 20.4° E | 0.42 |
| Zugspitze | 47.4° N | 11.0° E | 2.96 |
| Jungfraujoch | 46.5° N | 8.0° E | 3.58 |
| Izaña | 28.3° N | 16.5° W | 2.37 |
| Wollongong | 34.5° S | 150.9° E | 0.03 |
| Lauder | 45.0° S | 169.7° E | 0.37 |

**Table 2. Summary of correlation (R), mean bias and standard deviation of IASI-A and GOME-2A TOC products computed from daily gridded data, for each season of the period 2008 – 2017. The bias and the 1-σ standard deviation are given in percent. The correlation coefficients lower than 0.85 are indicated in italics.**

| Latitude range | Dec-Jan-Feb | | Mar-Apr-May | | Jun-Jul-Aug | | Sep-Oct-Nov | |
|---|---|---|---|---|---|---|---|---|
| | R | Bias (%) | R | Bias (%) | R | Bias (%) | R | Bias (%) |
| 90°S – 90°N | 0.96 | -1.3±4.5 | 0.98 | 0.4±4.1 | 0.97 | -0.8±3.8 | 0.93 | -0.7±3.4 |
| 60 – 90°N | 0.94 | -2.8±5.9 | 0.93 | -0.8±4.8 | 0.85 | -3.4±3.7 | 0.88 | -0.7±3.1 |
| 30 – 60° N | 0.96 | -3.0±3.8 | 0.97 | -1.3±3.6 | 0.93 | -1.2±3.3 | 0.90 | -1.3±2.8 |
| 0 – 30° N | *0.83* | -0.6±2.7 | 0.86 | 0.6±3.7 | *0.80* | 1.8±2.9 | *0.55* | 1.0±1.7 |
| 0 – 30°S | 0.86 | 0.2±2.5 | 0.82 | 1.1±2.3 | 0.89 | 2.0±2.5 | 0.87 | 0.9±2.5 |
| 30 – 60°S | 0.94 | -1.7±3.0 | 0.94 | -0.1±2.6 | 0.95 | -1.7±3.0 | 0.94 | -3.2±3.3 |
| 60 – 90°S | 0.94 | -1.1±3.4 | *0.62* | 3.5±3.9 | - | - | 0.94 | -2.1±5.2 |

**Table 3. IASI-A decadal trends and their 2-σ standard deviation (in %) calculated from the monthly relative differences between IASI and the FTIR data over the period 2008 – 2017 for the TOC and different partial ozone columns: surface-300 hPa (TROPO), 300-150 hPa (UTLS), 150-25 hPa (LMS) and 25-3 hPa (MS). The P-value is indicated into bracket. A P-value lower than 0.05 indicates a significant trend. Trends indicated in bold are significant.**

| | TROPO | UTLS | LMS | MS | TOC |
|---|---|---|---|---|---|
| Kiruna | **-16.1±8.1 (0.00)** | **-7.2 ±6.8 (0.03)** | **4.7±3.2 (0.00)** | 0.3±8.5 (0.96) | 0.10±2.4 (0.93) |
| Zugspitze | **-12.8±4.3 (0.00)** | **-10.5±5.8 (0.00)** | **-2.2±1.7 (0.01)** | 1.4±3.9 (0.48) | **-2.6±1.5 (0.00)** |
| Jungfraujoch | **-14.7±4.8 (0.00)** | **-11.2 ±6.2 (0.00)** | **-3.0±2.4 (0.02)** | 2.1±3.7 (0.27) | **-3.0±2.2 (0.01)** |
| Izaña | **-5.0±4.8 (0.04)** | **-7.1±5.9 (0.02)** | 0.2±2.0 (0.82) | **3.7±2.5 (0.00)** | 0.9±1.2 (0.14) |
| Wollongong | **-10.4±3.9 (0.00)** | 0.8±10.2 (0.89) | 0.6±1.8 (0.49) | 0.7±2.3 (0.53) | -0.5±1.0 (0.36) |
| Lauder | **-12.1±5.0 (0.00)** | **-8.2±6.1 (0.01)** | -0.0±1.6 (0.98) | 1.9±1.9 (0.05) | -0.8±1.1 (0.18) |

Figures

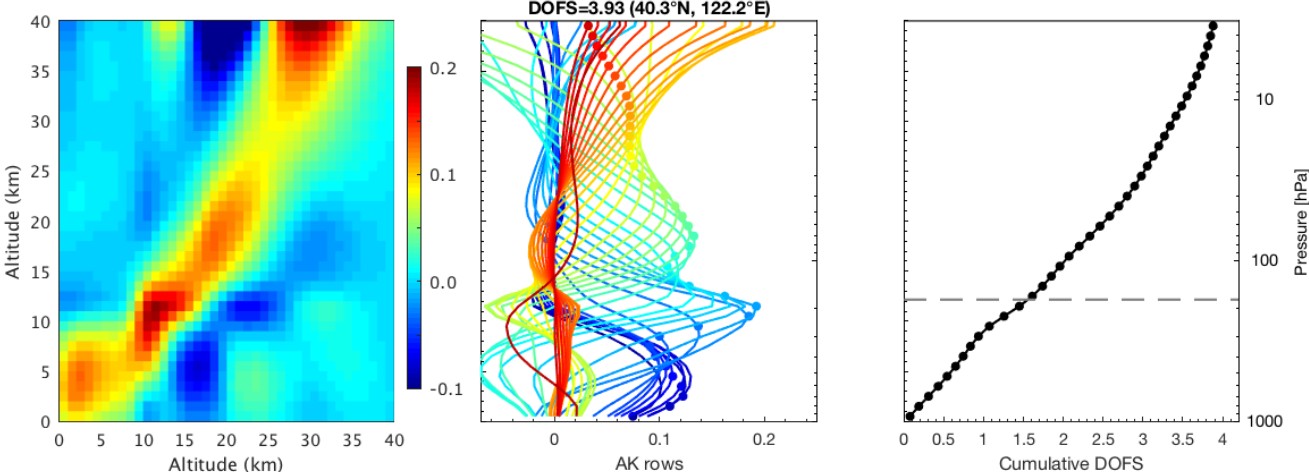

**Figure 1: (left) Example of the averaging kernel matrix for the IASI-A vertical profile retrieval indicating where the information present in the IASI-A vertical ozone profile (horizontal axis) originates from in the atmosphere (vertical axis). (middle) Other representation of the averaging kernel matrix (each line is a row of the averaging kernel matrix); The nominal height of each kernel is marked by a circle. (right) cumulative DOFS obtained from the diagonal of the averaging kernel matrix. The averaging kernels expressed in (molecules cm$^{-2}$) / (molecules cm$^{-2}$) correspond to one daytime mid-latitude measurement (40.3°N, 122.2°E) obtained on 1$^{rst}$ June 2016 for each 1 km retrieved layers from the surface to 40 k altitude.**

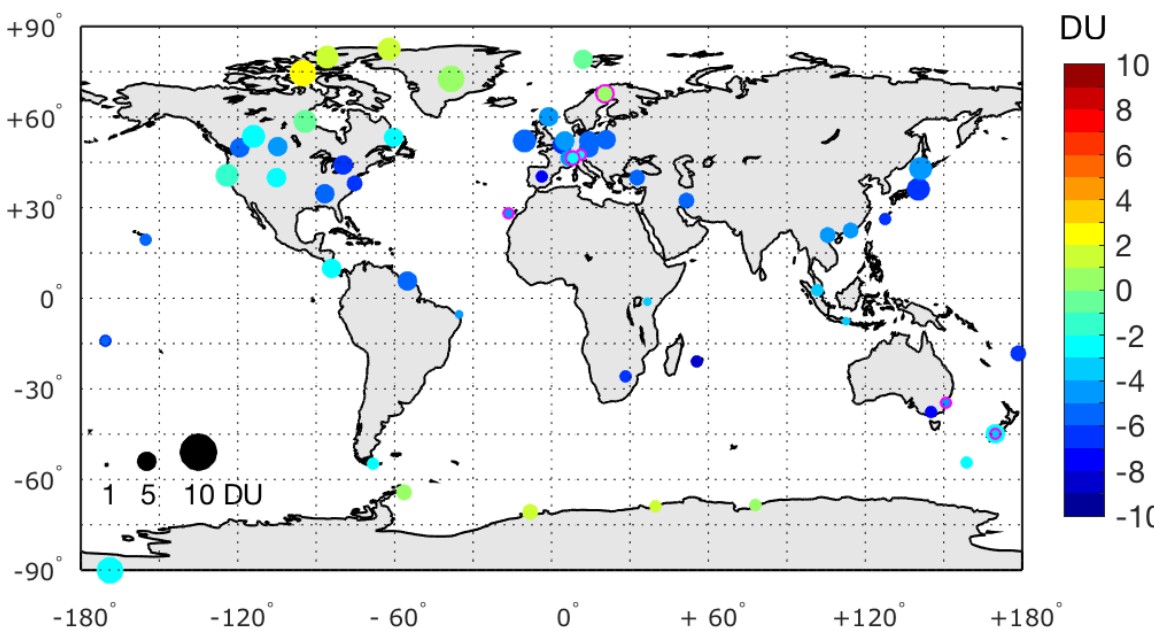

**Figure 2: Spatial distribution of ozonesonde and FTIR stations used in this study. The color represents the mean biases in Dobson units (DU) between IASI-A and sonde TROPO O$_3$ columns (as defined as the surface-300 hPa column) at each station and the dot size represents the standard deviation. The average is performed for the period January 2008 – July 2017. The mean bias between IASI-A and FTIR TROPO O$_3$ columns is indicated by the dots circled in magenta.**

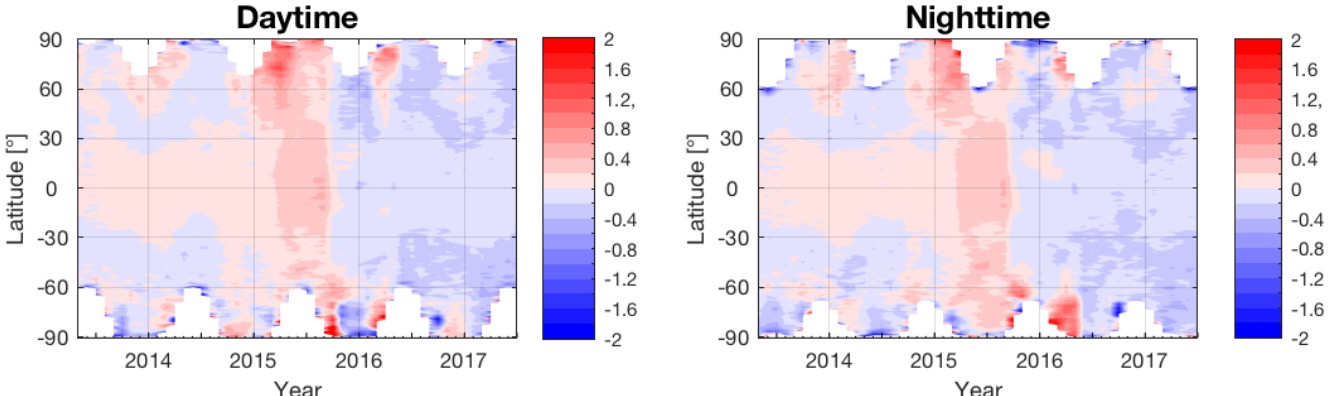

**Figure 3: Contour representation of the relative difference (in percent) between IASI-A and IASI-B Total Ozone Column (TOC) products for 1° zonal monthly mean TOCs for the period May 2013 – July 2017 for daytime data (left) and nighttime data (right). The relative differences are calculated as 100 x (IASI-A - IASI-B) / IASI-A.**

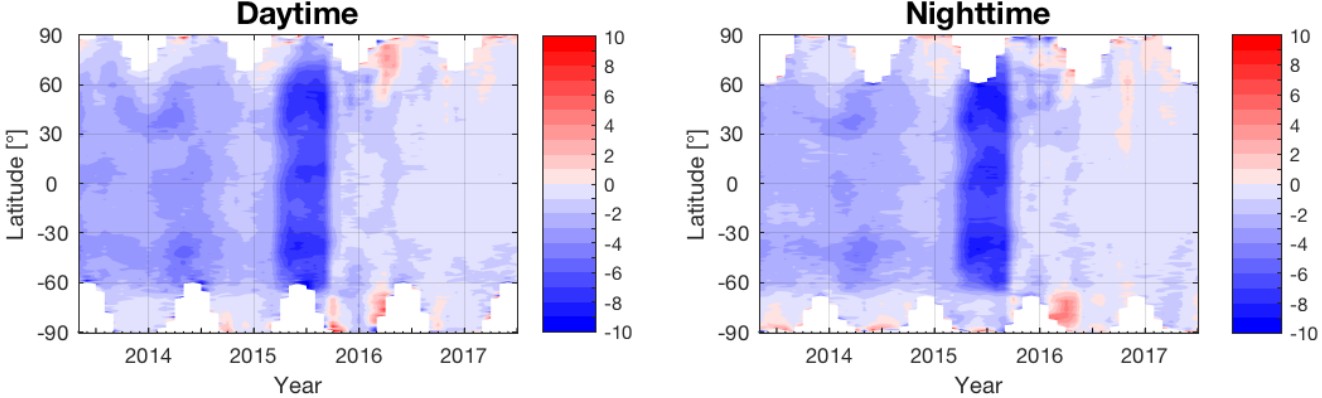

**Figure 4: Same as Fig. 3 for the TROPO $O_3$ column products (defined as the column integrated between the surface and 300 hPa).**

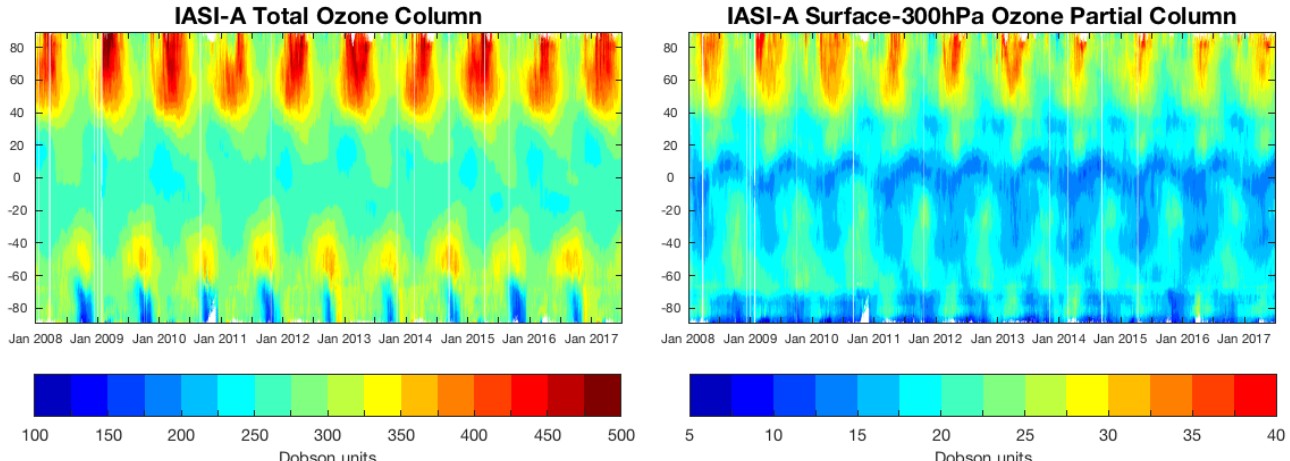

**Figure 5: IASI-A Total Ozone Column (left) and TROPO O₃ column (right) record (in Dobson units) as a function of latitude and time from January 2008 to July 2017. The TROPO O₃ column is calculated as the column integrated between the surface and 300 hPa.**

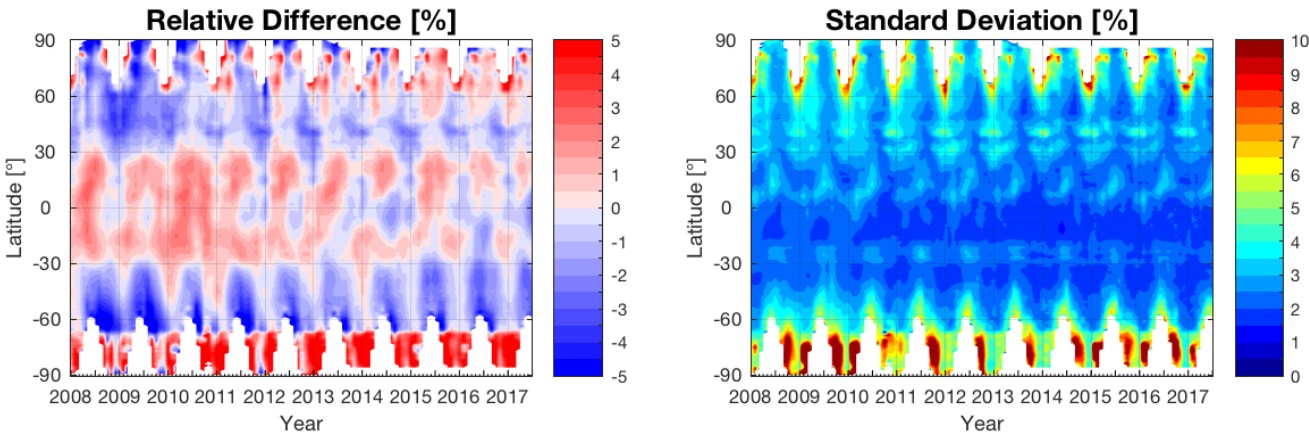

**Figure 6: (left) Relative differences (in percent) between IASI-A and GOME-2A for 1° zonal monthly mean Total Ozone Columns during the period 2008 – 2017; (right) Associated standard deviation (in percent). The relative difference is calculated as 100 x (IASI-A-GOME-2A) / GOME-2A.**

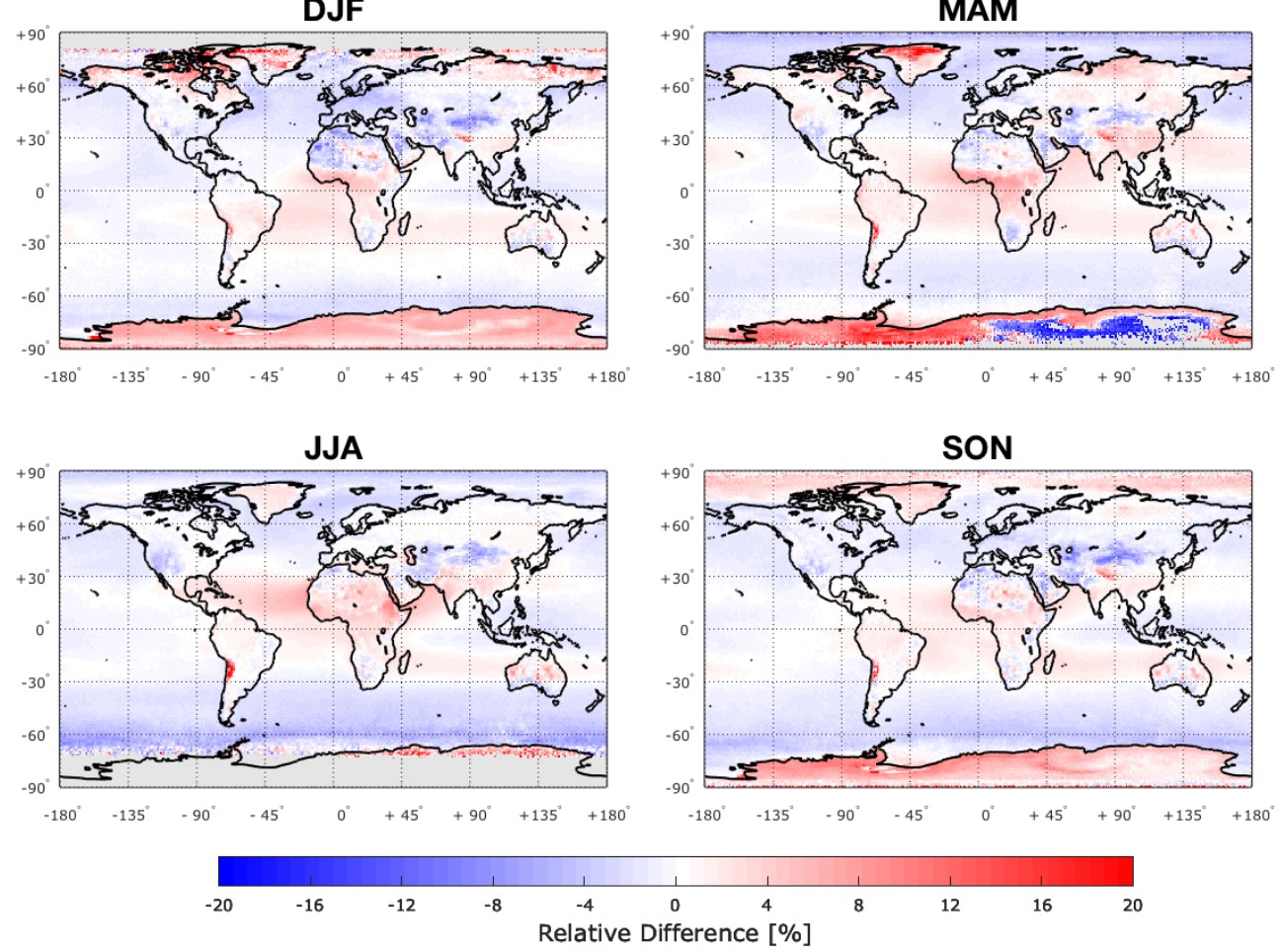

**Figure 7: Seasonal distribution of the relative differences (in percent) between IASI-A and GOME-2A Total Ozone Column products for the period 2008 – 2017. The relative difference is calculated as 100 x (IASI-A-GOME-2A) / GOME-2A.**

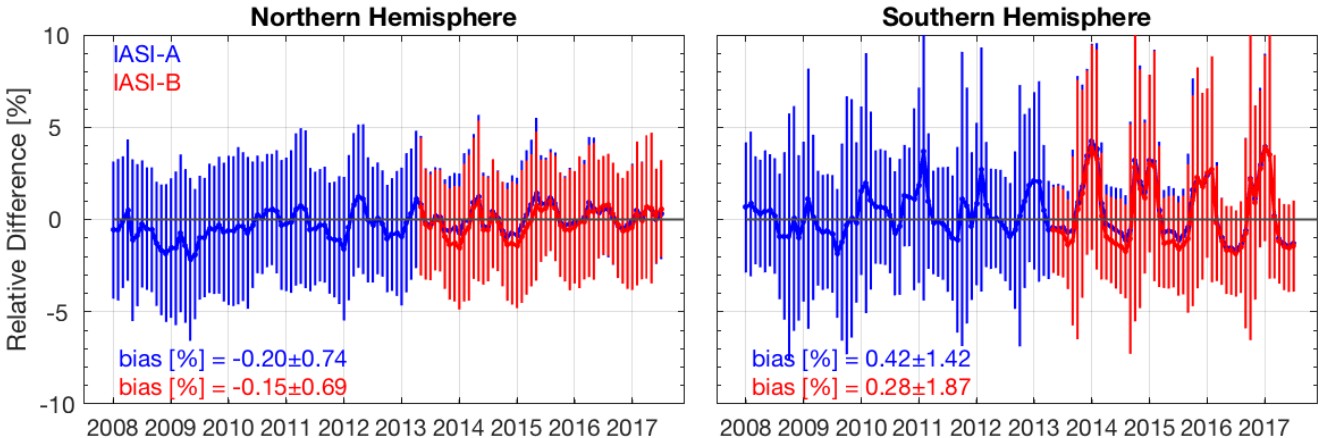

**Figure 8: Monthly relative differences (in percent) between IASI-A (blue) and IASI-B (red) against GOME-2A Total Ozone Column products as a function of time for the period 2008 – 2017 for the Northern Hemisphere (left) and the Southern Hemisphere (right). The 1-σ standard deviation of the relative differences is also displayed (vertical bars). For each 1°x1° grid cell, a relative difference is calculated as 100 x (IASI – GOME-2) / GOME-2 [%]. All the relative differences in each hemisphere are then monthly averaged. Comparison statistics including the mean bias and its 1-σ standard deviations in percent for the period 2008 – 2017 (IASI-A) and 2013 – 2017 (IASI-B) are indicated on each panel.**

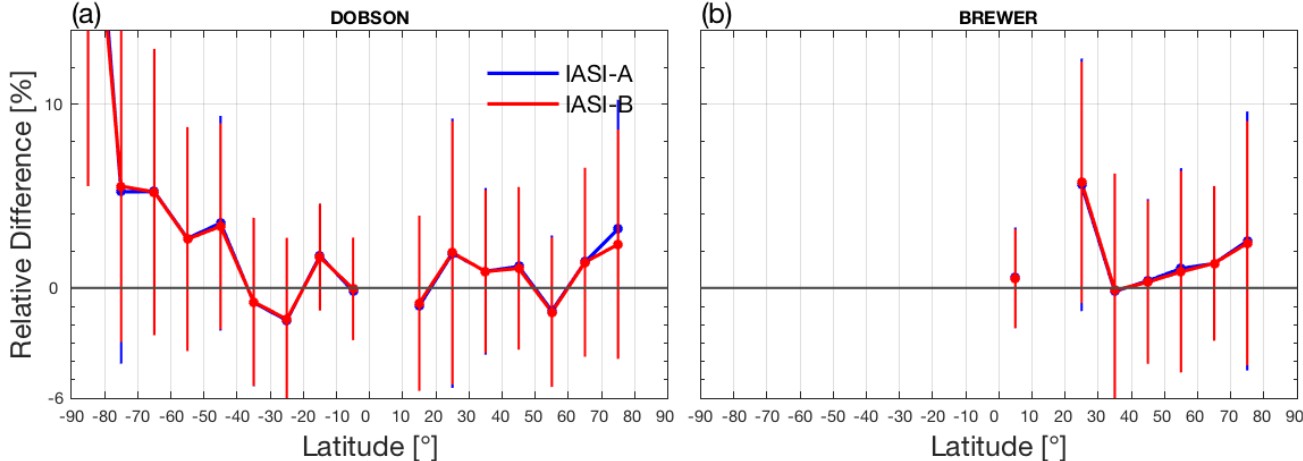

**Figure 9: Latitudinal variability of the relative difference (in percent) between IASI-A (blue) and IASI-B (red) against collocated Dobson (left) and Brewer (right) TOC data given in bins of 10°. Only the common collocations between the two satellites are shown (period May 2013 – July 2017). The 1-σ standard deviation of the relative differences is also displayed (vertical bars). The relative difference is calculated as 100 x (IASI – GB) / GB [%].**

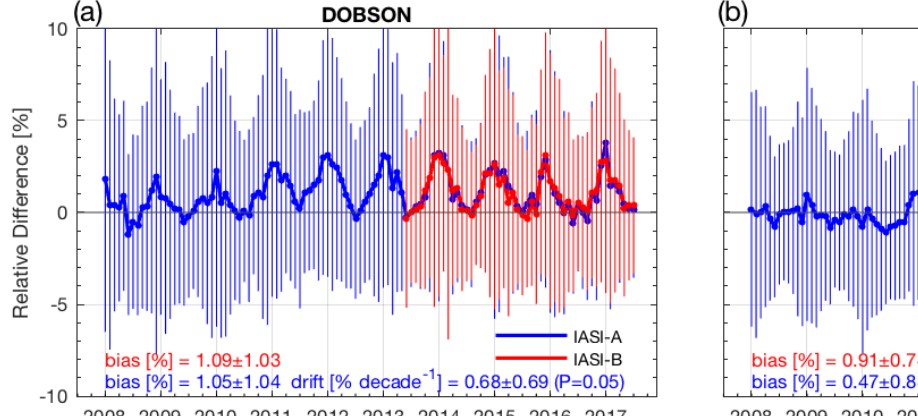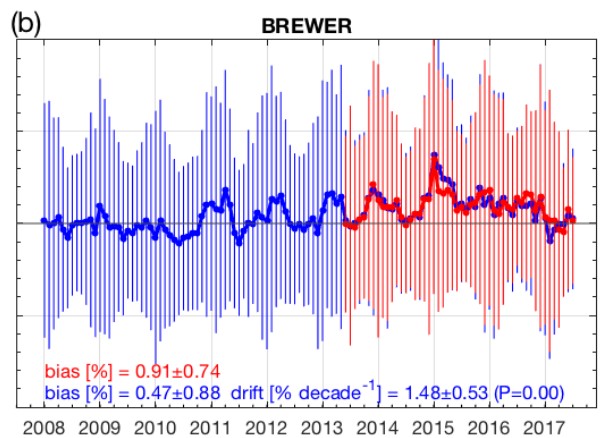

**Figure 10: Time series of the monthly relative differences (in percent) between IASI-A (blue) and IASI-B (red) against collocated ground-based (GB) TOC for the Northern Hemisphere for the Dobson network (left) and Brewer network (right). For each daily GB measurement, a relative difference is calculated as 100 x (IASI – GB) / GB [%]. All the relative differences are then monthly averaged. For the period May 2013 onwards, only the common collocations between IASI-A and IASI-B are shown. The 1-σ standard deviation of the average is also displayed (vertical bars). Comparison statistics including the mean bias and its 1-σ standard deviations in percent for the period 2008 – 2017 (IASI-A) and 2013 – 2017 (IASI-B) are indicated on each panel. The decadal drift in percent, its 2-σ standard deviation and the *P* value for the IASI-A time series are also indicated on each panel.**

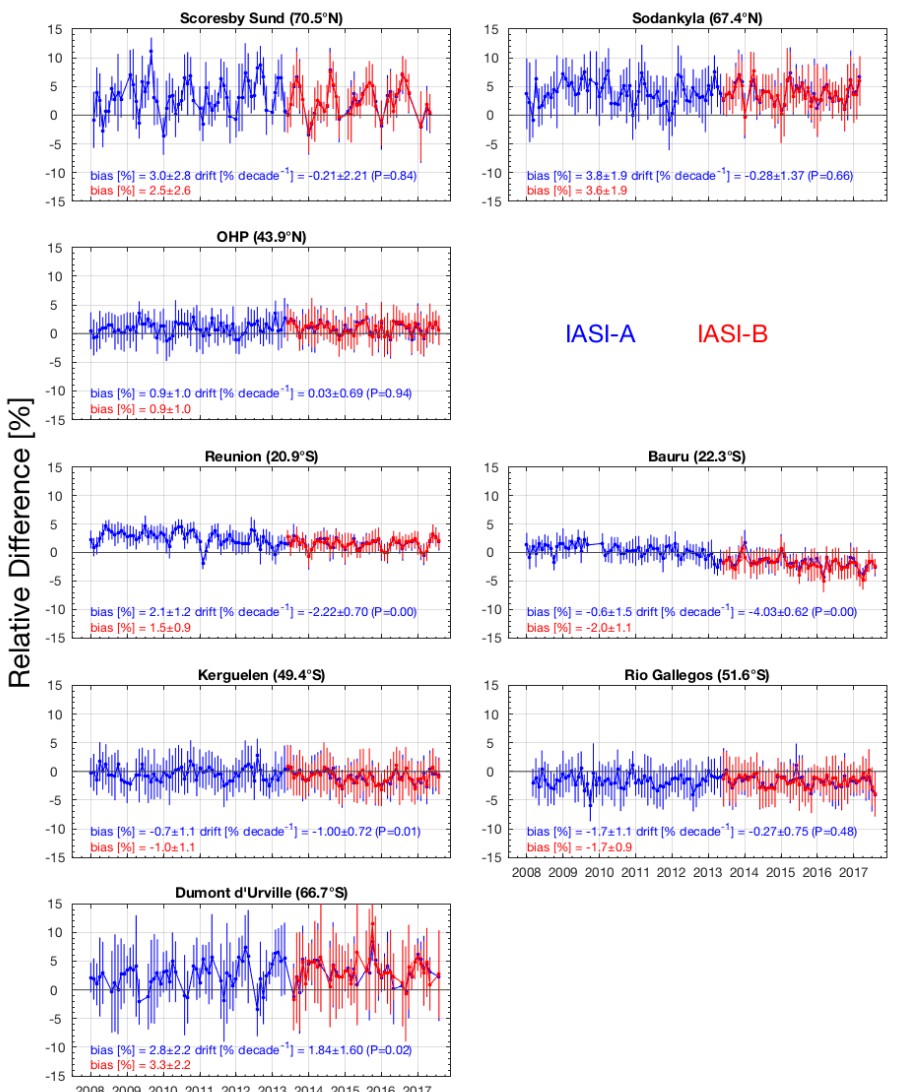

**Figure 11: Time series of the monthly relative differences (in percent) between IASI-A (blue) and IASI-B (red) against collocated SAOZ TOC measurements for eight stations from North to South. For each daily SAOZ measurement, a relative difference is calculated as 100 x (IASI – SAOZ) / SAOZ [%]. All the relative differences are then monthly averaged. For the period May 2013 onwards, only the common collocations between IASI-A and IASI-B are shown. The standard deviation of the average is also displayed (vertical bars). Comparison statistics including the mean bias and its 1-σ standard deviations in % for the period 2008 – 2017 (IASI-A) and 2013 – 2017 (IASI-B) are indicated on each panel. The decadal drift, its 2-σ standard deviation (in %) and the *P* value for the IASI-A time series are also indicated on each panel.**

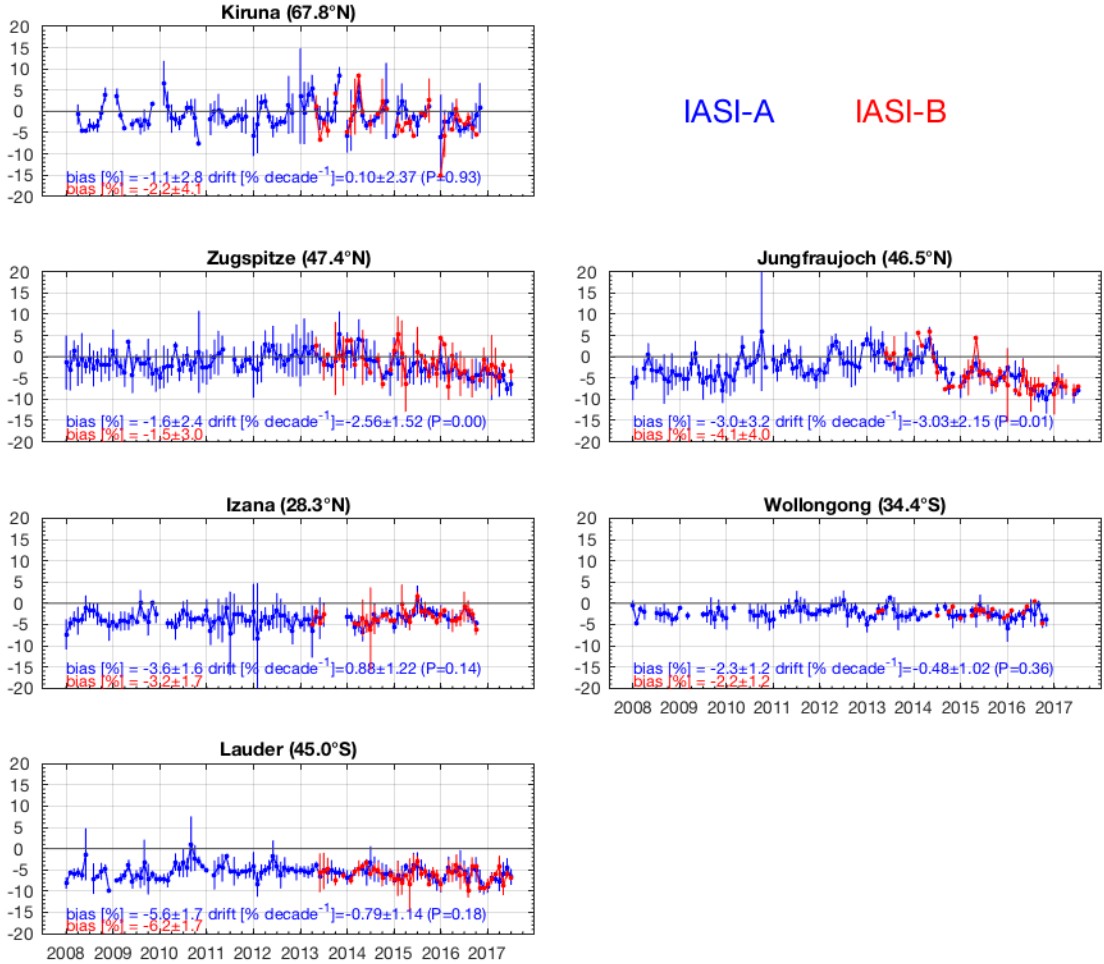

**Figure 12:** Time series of the monthly relative differences (in percent) between IASI-A (blue) and IASI-B (red) against collocated FTIR TOC measurements for six stations from North to South over the period 2008 – July 2017. For each daily FTIR measurement, a relative difference is calculated as 100 x (IASI – FTIR) / FTIR [%]. All the relative differences are then monthly averaged. The standard deviation of the average is also displayed (vertical bars). Comparison statistics including the mean bias and its 1-σ standard deviations in % for the period 2008 – 2017 (IASI-A) and 2013 – 2017 (IASI-B) are indicated on each panel. The decadal drift, its 2-σ standard deviation (in %) and the *P* value for the IASI-A time series are also indicated on each panel.

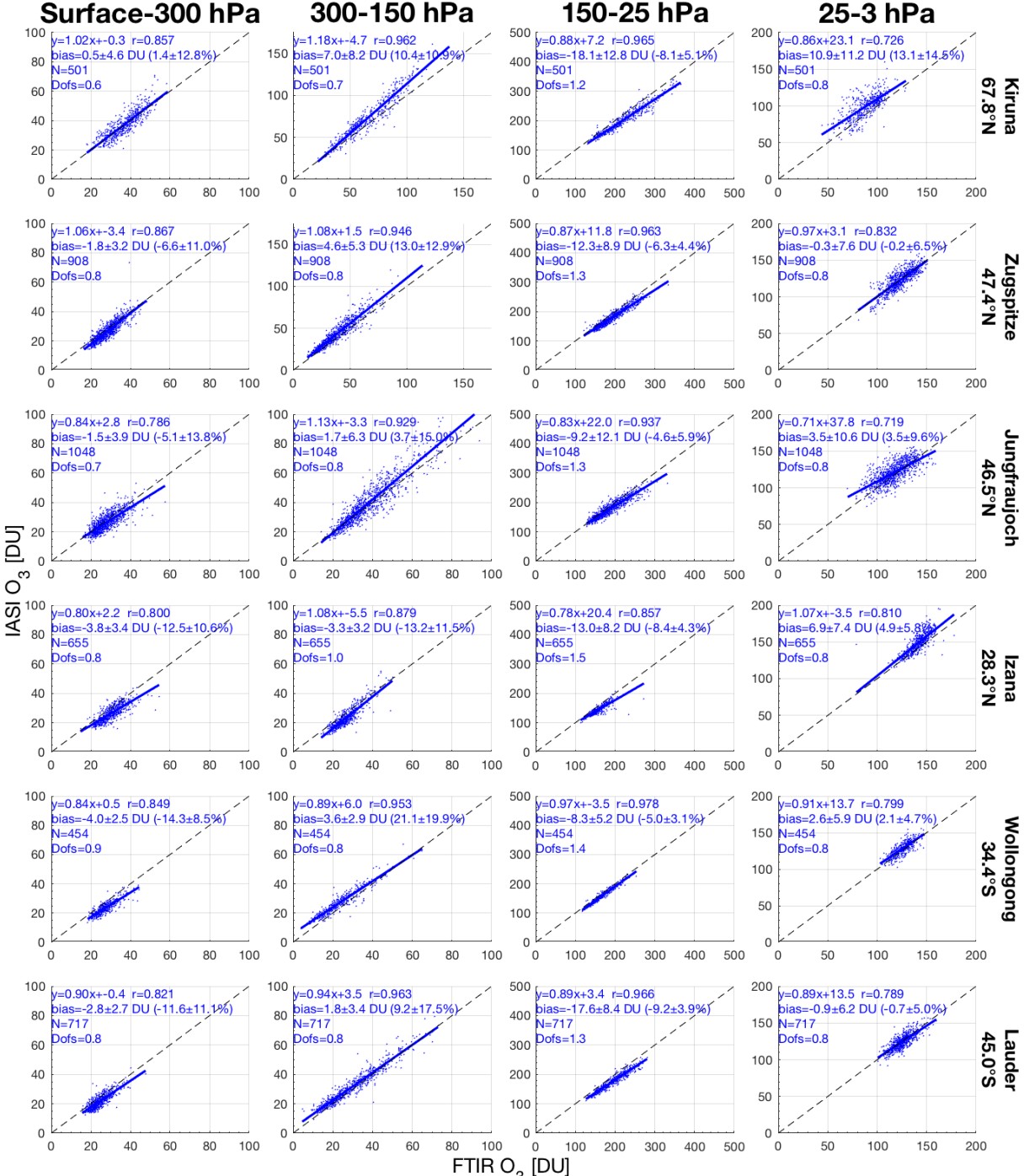

**Figure 13: Scatter plots of IASI-A against smoothed FTIR O₃ partial columns at six FTIR stations for the period 2008 –2017. Comparison statistics including the linear regression, the mean bias, its 1-σ standard deviation in both Dobson units (DU) and %, the number of collocations and the mean DOFS for each partial column are shown on each panel.**

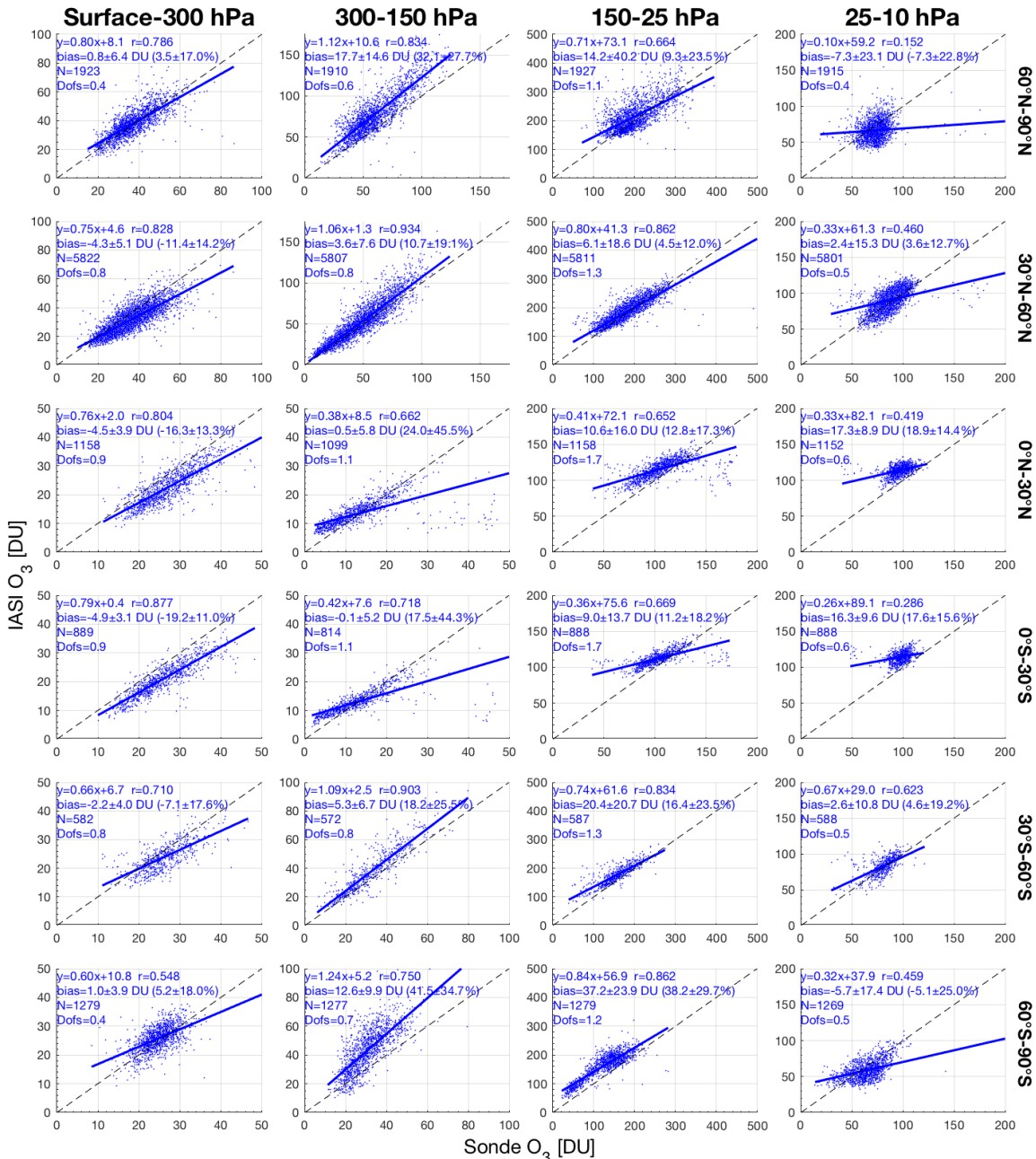

**Figure 14: Scatter plots of collocated IASI-A and smoothed ozonesonde O₃ partial columns for six latitude bands for the period 2008 – 2017. Comparison statistics including the linear regression, the mean bias, its 1-σ standard deviation in both Dobson units (DU) and %, the number of collocations and the mean DOFS for each partial column are shown on each panel.**

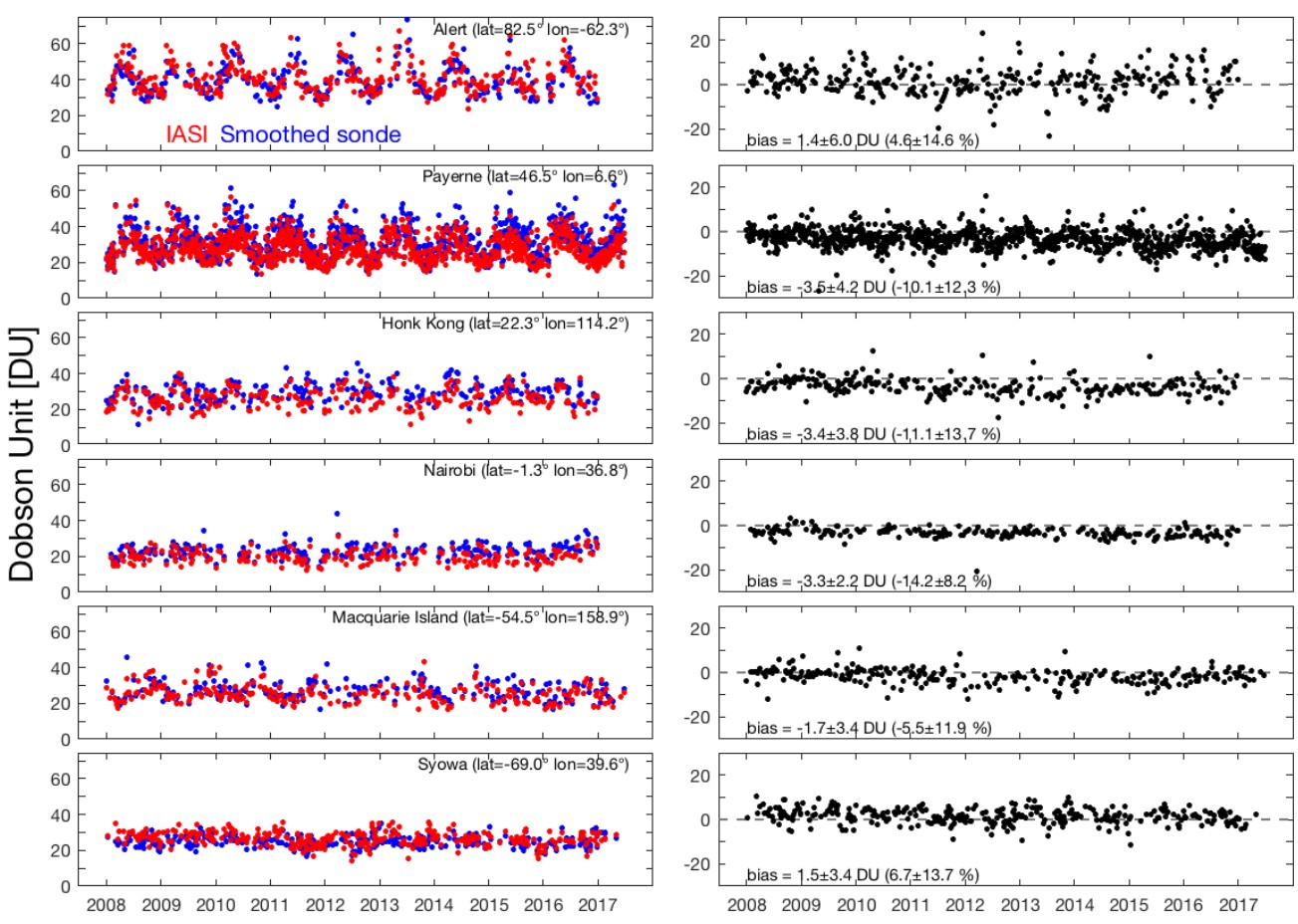

**Figure 15: (left panels)** Time series of daily IASI-A (in red) and smoothed ozonesonde (in blue) TROPO O₃ columns in Dobson Units (DU) for six stations representative of different latitude bands for the period 2008 – 2017; (right panels) Associated relative differences (in percent), calculated as 100 x ( IASI – SONDE) / SONDE, including the mean bias and its 1-σ standard deviation.

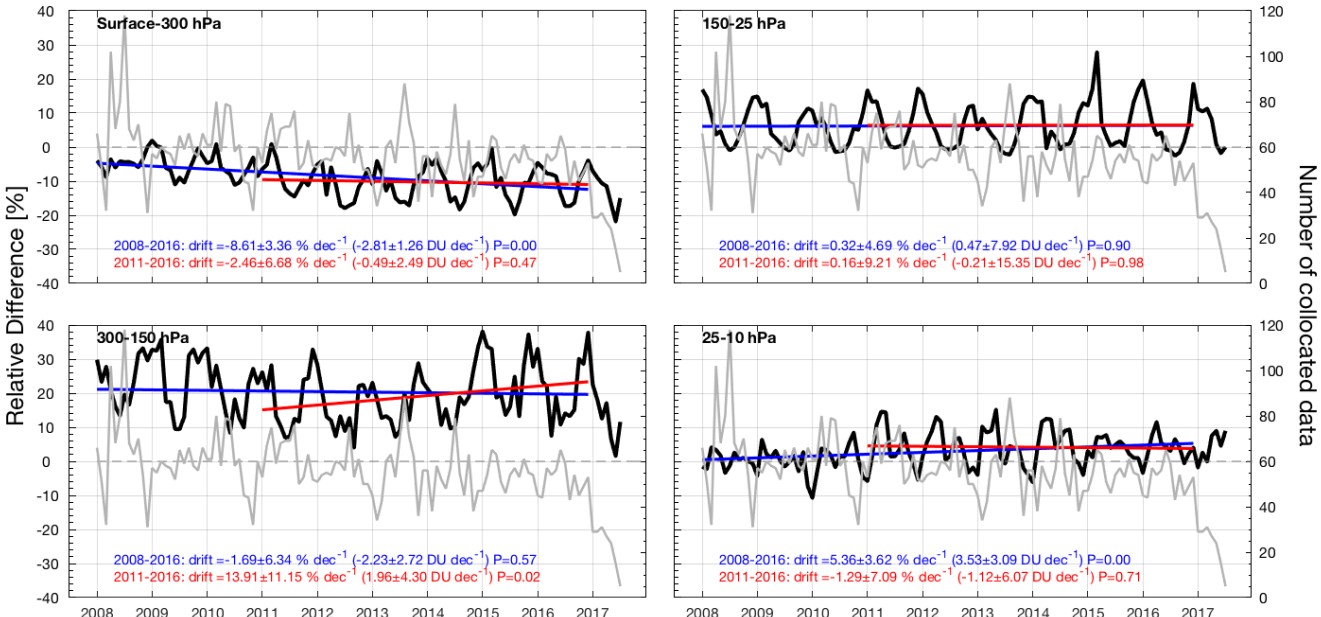

**Figure 16: Time series of the monthly mean relative differences between IASI-A and ozonesonde O₃ measurements for different partial columns for the period 2008 – 2017 for the Northern Hemisphere. The number of collocated data is also displayed in gray. The decadal drift in percent, its 2-σ standard deviation and the *P* value are indicated on each panel for two periods: 2008 – 2016 (blue) and 2011 – 2016 (red).**