# Peer review of "Validation of the IASI FORLI/Eumetsat ozone products using satellite (GOME-2), ground-based (Brewer-Dobson, SAOZ, FTIR) and ozonesonde measurements"

_Atmospheric Measurement Techniques, 2017_

## Referee Comment (RC1) · Anonymous Referee #1 · 16 Mar 2018

General: The article "Validation of the IASI FORLI/Eumetsat ozone products using satellite (GOME-2), ground-based (Brewer-Dobson, SAOZ) and ozonesonde measurements" submitted to AMT by A. Boynard et al. describes comparisons between IASI O3 retrieval data products and several other O3 data sets. They employ the FORLI version v20151001 retrievals. They used various time periods within the overall periods of observations IASI-A (2008-2017) and IASI-B (2013-2017) for overlap with other instrumental observations. The data have global coverage and is used in intercomparison between –A & -B. Other latitudinal or hemispheric comparisons are made with sparse

ground based observations. Comparisons are also made with total and partial column products.

Overall the article describes comparisons with 4 different datasets in a logical manner. There is considerable detail for any one of the comparisons that could be clarified better. That descriptions are brief may be necessary since there are several datasets to describe and there are references to previous work. Still the main points are not as forthcoming as they could/should be. There are no new techniques nor sophisticated procedures or concepts hence it should be clear where and especially why the comparisons are the state they are in.

This intercomparison is nearly identical to a previous comparison by the same author Boynard et al., 2016 yet not mentioned much. This current paper describes the latest FORLI version and the previous paper a previous FORLI version. But that hardly makes this new work, in fact many plots and tables are identical. This work should be cast as an update and a comparison to the previous FORLI version. In this way specific details on how the new version improves or changes O3 columns and partial column data are explicit. This is a large shortcoming of this submission and should be remedied before publication.

The IASI O3 retrieval is performed in the 1025-1075cm-1 IR region. Yet there are no comparisons with IR derived data sets. Such a comparison would diminish any discrepancy with cross section differences between IR and UV / Vis. This comparison would be seen as more thorough and results very interesting to take advantage of IR ground based datasets. Further, in particular NDACC IR data have vertical information comparable to IASI (DOFS $\sim$>4) to use for partial columns. Secondly, there is little discussion given to any contribution of cross section differences.

Specific: Throughout the text the adjectives 'good' or 'generally' are used in descriptions of a comparison. These qualitative comments do not help the reader nor are they appropriate. They are subjective and are detrimental to a real grasp of the state of

the IASI data with regard to other pertinent datasets. There are many uses of approximately ($\sim$) or less then ($<$) that seem inconsistent and hence then to obfuscate the real quality of the data.

Here are specific issues with the scientific points being made. 1. P1, L23 Brewer & Dobson TOC are not retrievals per se.

2. P1 L25 to wit "shows good long term stability" good relative to what?

3. P1 L 29 "Compared to ozonesonde data, IASI-A and IASI-B O3 products overestimate the O3 abundance in the stratosphere 30 (up to 20 % for the 150-25 hPa column) and underestimates the O3 abundance in the troposphere (within 10 % for the mid- latitudes and $\sim$18 % for the tropics). This sentence is needlessly confusing mixing zonal and altitude comparisons and using hPa layers an "troposphere".

4. P1 L32 "small" compared to what?

5. P2 L21 "180 shift" is not clear. It's a shift in what?

6. P4 L24 what is a O3 profile "C-shape"

7. P5 L 9 " Differences between IASI-A and IASAI-B. Is the plot A-B or B-A ? More generally for all comparisons in this paper most are ambiguous on this simple point. Every instance in the text and captions should be made explicit.

8. P6 L2 'excellent agreement' How is excellent agreement defined? Is there a reference for comparison of spectra?

9. P7 L21-27 Here are given possible sources of differences at high latitudes between IASI & GOME data. They are apparently (per reference) the same or very similar to Boynard 2016. These do not help the reader know if and/or how FORLI v20151001 is an improvement. For instance given GOME data quality and known issues with UV/Vis retrievals from GOME, what changes in v20151001 make it an improvement over v20140922? More specifically i) 'limited information content. . .low surface temperatures'. Is this due to lower spectral SNR? Or something else? ii) misrepresentation of surface emissivity is vague. Are these values have large errors? Or have large biases or both? Is there a reference? iii)if the temperature profiles are known to be less reliable this implies variability not necessarily bias (but this list is describing a GOME/IASI bias. Further what magnitude temperature error delivers the O3 bias seen?

10. P8 L1 Is there some explanation what the physical basis is for the rejection of data over deserts and Antarctica?

11. P11 L18 Its not completely clear are all comparisons of the upper most partial column to 10hPa despite the statement to the contrary on L17 (25-3hPa)?

12. P11 L22 What is the source of the extremely low O3 in the UTL? Is it low in the sonde, IASI or both? How much data is lost?

13. P12 L8 It is very reasonable to follow that the high variability in the UTLS give large SD but less so to see what the a priori has to contribute given DOFS ∼1.

14. P12 L14 Its not clear what information is missing? The comparison is to 10hPa for both instruments – is that correct?

15. P12 L29 Please clarify are these 40 pairs globally?

16. P13 L21 Please provide a reference for the stability of the IASI radiances.

17. P13 L31 use of the adjective generally is not helpful.

Technical: 1. P1 L19 what does ir "generally consistent" versus just "consistent"

2. P1 L23 "retrieved ones" would better be "retrieved TOC's"

3. P1 L24 "on the instruments" would better be "on the compared instruments"

4. P1 L30 (up to 20 % for the 150-25 hPa column) should be (up to 20 % for the 150-25 hPa partial column)

5. P1 L30 "within" might better be "less then"

6. P2 L 4 "amount" should be "amounts"

7. P9 L2 'latitude belt' might be better worded 'latitude band'

8. P3 L11 "overestimates the ultraviolet (UV) Total Ozone Column (TOC)" Do you really mean to differentiate the uv TOC from some other spectral region?

9. P5 L19 'posteriori' should be 'after'

10. P5 L 31 remove second 'between April...'

11. P6 L1 'proved' might better be worded 'shown'. Fruthermore is there a reference for this statement/conclusion?

12. P6 L13 'being preferentially be used' might better be worded 'are recommended'.

13. P12 L20 From plot 14 I read +1.5 & − 3.5 % difference (not +- 3.5) and SD of maximum 14.6% (not 14)

14. P12 L 23 Dobson units (sp)

15. P12 L28 ...O3 partial columns...

16. P13 L20 ...proven to be very...

17. P14 L7 'better from October' might be better worded 'better after October'

18. P33 caption, use of the term 'sub' column should be removed (in all cases) and partial be used for consistency.

---

## Referee Comment (RC2) · Anonymous Referee #2 · 20 Mar 2018

Review of "Validation of the IASI FORLI/Eumetsat ozone products using satellite (GOME-2), ground-based (Brewer-Dobson, SAOZ) and ozonesonde measurements"

General comments

The manuscript titled "Validation of the IASI FORLI/Eumetsat ozone products using satellite (GOME-2), ground-based (Brewer-Dobson, SAOZ) and ozonesonde measurements" give a thorough validation study of 9 years of IASI ozone measurement. The manuscript is well written, clear and easy to read.

[Figure]

However, it is not easy to understand whether this paper (Boynard et al., 2017) presents novel concepts, ideas, tools or data, especially when we compare Boynard et al. (2017) to Boynard et al. (2016). Many conclusions in Boynard et al. (2016) and in Boynard et al. (2017) are similar.

Were the authors expecting different results between IASI v20140922 and v20151001? According to Boynard et al. (2016), the improvement of IASI retrieval was already found to be mainly located in the middle stratosphere.

How much could the bias assessment change with 2 more years of data?

9 years of data allow the authors to address the long-term stability of IASI. This is the most interesting and the newest part of the study. Unfortunately the significant drift in the troposphere is barely explained and addressed.

Furthermore, Boynard et al. (2017) didn't address open questions already mentioned in the conclusions of Boynard et al. (2016), such as the large bias found in the UTLS, still not fully understood.

For these reasons, I would suggest major corrections before the current manuscript can be published in AMT.

Specific comment:

- Section 3: Intercomparison between IASI-A and IASI-B ozone products
Line 18 p. 5: Change "the figure" to "Figure 2"

Line 2 p. 6: Change "then" to "October 2015"

Line 14 p. 6: April-October 2015 shouldn't be included in the combined IASI-A/IASI-B product (as explained in Line 10), because of instrumental issues on IASI-A. Should this time-period be excluded from any time-series studies with IASI-A?

- Section 4: Validation of IASI-A and IASI-B total ozone columns
I would suggest to move all the validation method in one method section. The method section could then include: (1) the formulae of differences calculation, (2) The method of co-location between IASI and reference observations, (3) the characteristics of the

data used for the comparison. This change would help the authors to shorten several sub-sections.

Line 25-27 p. 7: This statement is almost word to word the same as in Boynard et al. (2016). Don't the author think that "Further investigation would be needed to understand the reasons of these larger differences at high latitude" should be addressed in the current study?

Line 33 p. 7: Would it be possible to quantify the "better agreement"?

Lines 3-11 p. 8: All this paragraph is already stated in Boynard et al. (2016). It could be either removed or shortened.

Line 10 p. 8: Would it be possible to quantify the "magnitude"?

Line 28 p. 8: Could you explain why the number of stations would influence the dependency on the latitude of the differences between IASI and GB measurements?

Line 29 p. 8: The differences between IASI-A and Dobson seem to reach 3.5% in NH, while the authors report [0-2.5%]

Line 30 p. 8: "Lower than 40°S" would mean somewhere between 0 and 40°S. Do you mean between 40°S and 60°S? Please clarify.

Line 1 p. 9: It is worth to notice there is no Brewer measurements in SH.

Line 2 p. 9: Change "belt" to "region'.

Lines 5-6 p. 9: Could you explicitly mention the 1-3% requirement from the Ozone_cci project instead of " within $\pm 3\%$".

Line 11 p. 9: Change "small" to " < 3% ".

Lines 12-13 p. 9: In the (new) method section I would suggest to explain the ozone-cci project and their requirement in term of satellite products stability. Could you explain how the 1-3% requirement has been decided? According to the 1-3% requirement,

"IASI-A TOC products are reliable for trend studies". Does it mean no drift adjustment at all is required? And does it mean that the drift, even small, is not taken into account in the ozone trends uncertainties? Could you please explain?

Line 13 p. 9: Which criteria is used to qualify differences "within 1.1%" as "very good agreement"?

Lines 16-20 p. 9: This paragraph is not clear. It is hard to understand what would explain differences in the seasonal variability between Dobson and Brewer. What does 0.6% represent?

- Section 5: Validation of IASI-A and IASI-B partial ozone column products
As mention for Section 4, I would suggest to move the comparison method in one method section.

Line 16 p.10: Could you report the numbers of the "small or non-significant negative decadal trends"?

Lines 16-17 p.10: Could you refer again to the 1-3% requirement with the reference of the Ozone_cci project?

Line 27 p. 10: "[. . .] their uncertainties are lower than other types of ozonesondes [. . .]" Could you quantify?

Lines 6-12 p. 11: The common method to compare satellite data with ozonesondes is to degrade the high vertically resolved ozonesondes by applying the AKs and a priori ozone profiles used to retrieve satellite ozone products. In Huang et al. (2017), they use the high vertically resolved ozonesondes (without degrading the vertical resolution) in the regions and altitudes when the satellite has low retrieval sensitivity.
Could you comment on this? Is such analysis could be done in your study?
Huang, G., Liu, X., Chance, K., Yang, K., Bhartia, P. K., Cai, Z., Allaart, M., Ancellet, G., Calpini, B., Coetzee, G. J. R., Cuevas-Agulló, E., Cupeiro, M., De Backer, H., Dubey, M. K., Fuelberg, H. E., Fujiwara, M., Godin-Beekmann, S., Hall, T. J., Johnson,

B., Joseph, E., Kivi, R., Kois, B., Komala, N., König-Langlo, G., Laneve, G., Leblanc, T., Marchand, M., Minschwaner, K. R., Morris, G., Newchurch, M. J., Ogino, S. Y., Ohkawara, N., Piters, A. J. M., Posny, F., Querel, R., Scheele, R., Schmidlin, F. J., Schnell, R. C., Schrems, O., Selkirk, H., Shiotani, M., Skrivánková, P., Stübi, R., Taha, G., Tarasick, D. W., Thompson, A. M., Thouret, V., Tully, M. B., Van Malderen, R., Vömel, H., von der Gathen, P., Witte, J. C., and Yela, M.: Validation of 10-year SAO OMI Ozone Profile (PROFOZ) product using ozonesonde observations, Atmos. Meas. Tech., 10, 2455-2475, 10.5194/amt-10-2455-2017, 2017.

Lines 18 and 31 p. 12: The selection of the ozonesondes stations are confusing. Why don't you use all the ozonesondes stations that meet the criteria needed for the comparison such as long-term time series, statistics of the data, etc...?

Lines 14-24 p. 13: This part of the discussion is one of the most interesting but it is too short. Would it be possible to address at least one of the speculative explanation for such a drift?

- Summary
Line 4 p. 14: Would you suggest to remove the data between April and September 2015 (October 2015 in the main text) for trends studies? If so, could you mention it? Would it be possible to apply any corrections factor on the data for this time-period?

Line 12-14 p. 14: What do you mean by "due to larger differences at the southern high latitudes"? The sentence is not clear.

Line 20 p. 14: Could you report the numbers for "insignificant negative trends"? Do you refer to the P-value for "insignificant"?

Line 25 p. 14: The statement about the large biases found in the UTLS was already mentioned in Boynard et al. (2016), but still it is not fully understood. Could you address this question in your study?

---

## Author Comment (AC1) · 1 Jun 2018

**Response to Referee #1**

**The authors would like to thank the referee for her/his general comments about the manuscript and her/his useful suggestions and corrections, which have helped us clarifying several points and improving the manuscript. Below are our responses to the comments brought up by the referee. The referee's comments and our responses are marked in blue and in black, respectively. In italic are the changes made in the manuscript.**

General: The article "Validation of the IASI FORLI/Eumetsat ozone products using satellite (GOME-2), ground-based (Brewer-Dobson, SAOZ) and ozonesonde measurements" submitted to AMT by A. Boynard et al. describes comparisons between IASIO3 retrieval data products and several other O3 data sets. They employ the FORLI version v20151001 retrievals. They used various time periods within the overall periods of observations IASI-A (2008-2017) and IASI-B (2013-2017) for overlap with other instrumental observations. The data have global coverage and is used in intercomparison between –A & -B. Other latitudinal or hemispheric comparisons are made with sparse ground based observations. Comparisons are also made with total and partial column products.

Overall the article describes comparisons with 4 different datasets in a logical manner. There is considerable detail for any one of the comparisons that could be clarified better. That descriptions are brief may be necessary since there are several datasets to describe and there are references to previous work. Still the main points are not as forthcoming as they could/should be. There are no new techniques nor sophisticated procedures or concepts hence it should be clear where and especially why the comparisons are the state they are in. This intercomparison is nearly identical to a previous comparison by the same author Boynard et al., 2016 yet not mentioned much. This current paper describes the latest FORLI version and the previous paper a previous FORLI version. But that hardly makes this new work, in fact many plots and tables are identical. This work should be cast as an update and a comparison to the previous FORLI version. In this way specific details on how the new version improves or changes O3 columns and partial column data are explicit. This is a large shortcoming of this submission and should be remedied before publication.

In order to take into account Reviewer #1's main concern, we better quantify the improvements of the new version of FORLI in comparison with the previous one in the revised manuscript. The improvement for ozone partial columns is also discussed. We also highlight the fact that the improvement is rather constant over the globe and therefore issues are still persisting over some regions such as high latitudes, mountain region and desert. Specific studies have been initiated with different validation groups in order to assess the reasons for the larger differences and will be the object of an independent study. Here are the changes made in the revised manuscript:

GOME-2 comparison section:
"*Globally, IASI-A (IASI-B) TOC product are slightly higher than GOME-2A TOC product, with a global mean bias of 0.3±0.8 % (0.4±0.8 %). It is worth noting that the previous IASI TOC product (v20140922) was in disagreement by more than 5 % (Boynard et al., 2016). The global mean bias is now within total errors of GOME-2 estimated to 3-7 % (Valks et al., 2017) and IASI, which demonstrates the good consistency between IASI and GOME-2 TOC products.*"

Brewer/Dobson comparison section:
"*Nevertheless the overall comparison with Dobson and Brewer TOCs shows that IASI new TOC product is improved by 4% in comparison with the previous IASI TOC product (v20140922; see Boynard et al. (2016)) and is within IASI and GB TOC total error bars.*"

SAOZ comparison section:

"*The results are consistent with those found for the comparison with GOME-2A along with Brewer and Dobson measurements (see Sections 5.1 and 5.2, respectively). An improvement of 3-4 % is found when compared to the previous IASI product (v20140922).*"

Ozonesonde comparison section:

"*In comparison with the previous IASI partial ozone column products reported in Boynard et al. (2016), the new IASI ozone product is significantly improved in the MS by 8-12 % for the mid latitudes and tropics. The improvement is less significant for the LMS except in Antarctic where an improvement of 6 % is found. As for the TROPO and UTLS columns, no or slight improvement (<2 %) is found, and the agreement between IASI and sonde data is even worse compared to the previous IASI ozone product, especially for the southern tropical TROPO column (by 7 %) and the UTLS column (by 10-18 %)*"

The IASI O3 retrieval is performed in the 1025-1075cm-1 IR region. Yet there are no comparisons with IR derived data sets. Such a comparison would diminish any discrepancy with cross section differences between IR and UV / Vis. This comparison would be seen as more thorough and results very interesting to take advantage of IR ground based datasets. Further, in particular NDACC IR data have vertical information comparable to IASI (DOFS ~>4) to use for partial columns. Secondly, there is little discussion given to any contribution of cross section differences.

We thank the referee for his/her suggestion of adding a comparison with IR remote sensed data. We followed his/her suggestion by comparing IASI and FTIR TOCs and partial ozone columns for several FTIR stations. As for the discussion about any contribution of cross section differences, we discuss it in the new IASI/FTIR section. Here are the changes made in the revised manuscript regarding the new IASI/FTIR comparison :

[revised manuscript text omitted]

Specific: Throughout the text the adjectives 'good' or 'generally' are used in descriptions of a comparison. These qualitative comments do not help the reader nor are they appropriate. They are subjective and are detrimental to a real grasp of the state of the IASI data with regard to other pertinent datasets. There are many uses of approximately (~) or less then (<) that seem inconsistent and hence then to obfuscate the real quality of the data.
We removed the adjectives 'good', 'generally', ~ and < as much as possible.

Here are specific issues with the scientific points being made.
1. P1, L23 Brewer & Dobson TOC are not retrievals per se.
We removed the word 'retrieved'.

2. P1 L25 to wit "shows good long term stability" good relative to what?
We removed the word 'good'.

3. P1 L 29 "Compared to ozonesonde data, IASI-A and IASI-B O3 products overestimate the O3 abundance in the stratosphere 30 (up to 20 % for the 150-25 hPa column) and underestimates the O3 abundance in the troposphere (within 10 % for the mid- latitudes and ~18 % for the tropics). This sentence is needlessly confusing mixing zonal and altitude comparisons and using hPa layers an "troposphere".
We changed the sentence to:
"*Compared to ozonesonde data, IASI-A and IASI-B $O_3$ TROPO column (defined as the column between the surface and 300 hPa) is positively biased in the high latitudes (4-5 %) and negatively biased in the mid-latitudes and tropics (11-13 % and 16-19 %, respectively).*"

4. P1 L32 "small" compared to what?
We removed the word "small".

5. P2 L21 "180 shift" is not clear. It's a shift in what?
Metop-A and Metop-B satellites are 180° out of phase and thus for one specific location one satellite may be before or after the other. We made this part clearer as follows:

"*The two Metop satellites are on the same orbit with Equator crossing times of 09:30 (21:30) local mean time solar time for the descending (ascending) part of the orbit. There are therefore numerous common observations between two consecutive tracks. However, since Metop-A and Metop-B are 180° out of phase, there is a ~50 min temporal difference between both instruments (one satellite might be before or after the other); thus the observations are never quite simultaneous.*"

6. P4 L24 what is a O3 profile "C-shape"
A C-shape $O_3$ profile is a profile characterized by an abnormal increase in $O_3$ at the surface, as shown in the figure below. We rephrased the sentence in the revised manuscript as follows:
"(vii) the $O_3$ profiles have an unrealistic C-shape *(i.e. abnormal increase in $O_3$ at the surface, e.g. over desert due to emissivity issue)*, with a ratio of the surface – 6 km column to the total column higher or equal to 0.085;"

[Figure]

Figure: Example of IASI vertical profiles for one day. The red profiles correspond to C-shape profiles.

7. P5 L 9 " Differences between IASI-A and IASI-B. Is the plot A-B or B-A ? More generally for all comparisons in this paper most are ambiguous on this simple point. Every instance in the text and captions should be made explicit.

The relative difference between IASI-A and IASI-B is calculated as: 100 x (IASIA – IASIB) / IASIA. We made it clearer in the manuscript as follows:

"*Before validating IASI-A and IASI-B O$_3$ products, we assess the consistency between both instruments over the common period May 2013 – July 2017. For the intercomparison exercise, we first calculate the daily IASI-A and IASI-B averages over a 1°x1° grid. Then for each 1°x1° grid cell, we calculate the relative difference as 100x[(IASI-A – IASI-B)/IASI-B]. Finally we calculate the monthly averaged data from the daily gridded differences. A statistical analysis of IASI-A and IASI-B TOCs and TROPO O$_3$ columns is performed with respect to time and latitude.*"

The formulae of the difference calculation is also indicated in the revised captions.

Following a comment from Referee #2, we have moved the dataset characteristics and validation method in two sections:
- **2. IASI measurements and datasets used for the validation**: this section describes the IASI $O_3$ retrievals as well as the independent measurements used for the validation
- **3. Comparison methodology**: this section includes the formulae of difference calculation as well as the different comparison methods used in the present work.

8. P6 L2 'excellent agreement' How is excellent agreement defined? Is there a reference for comparison of spectra?
We changed 'excellent agreement' to 'better quality' and we rephrased the sentence as follows:
"*As a result, since October 2015, the IASI-A and IASI-B spectra are of better quality/stability  (Buffet et al., 2016; Jacquette et al., 2016).*"

Buffet, L., Villaret, C., Jacquette, E., Vandermarcq, O., Astruc, P., and Anstötz, S. :Status of IASI instruments onboard Metop-A and Metop-B satellites, 4th IASI International Conference, Antibes Juan-Les-Pins, France, 11-15 April 2016, https://iasi.cnes.fr/sites/default/files/drupal/201612/default/bpc_iasi-conference4-1_02_instruments_buffet.pdf, 2016.

Jacquette, E., Maraldi, C., Standfuss, C., Coppens, D., Delatte, B., Baqué, C., Calvel, J.-C., Buffet, L., Vandermarcq, O. : IASI performance assessment after permanent cube corner compensation device stop, 4th IASI International Conference, Antibes Juan-Les-Pins, France, 11-15 April 2016, https://iasi.cnes.fr/sites/default/files/drupal/201612/default/bpc_iasi-conference4-s1-08_jacquette.pdf, 2016.

9. P7 L21-27 Here are given possible sources of differences at high latitudes between IASI & GOME data. They are apparently (per reference) the same or very similar to Boynard 2016. These do not help the reader know if and/or how FORLI v20151001 is an improvement. For instance given GOME data quality and known issues with UV/Vis retrievals from GOME, what changes in v20151001 make it an improvement over v20140922? More specifically i) 'limited information content. . .low surface temperatures'. Is this due to lower spectral SNR? Or something else? ii) misrepresentation of surface emissivity is vague. Are these values have large errors? Or have large biases or both? Is there a reference? iii) if the temperature profiles are known to be less reliable this implies variability not necessarily bias (but this list is describing a GOME/IASI bias. Further what magnitude temperature error delivers the O3 bias seen?

In Boynard et al. (2016) we identified a systematic bias between the IASI data and observations obtained using the UV spectral range. The amplitude of the bias varies in latitude, and is more pronounced at higher latitudes. In v20151001 we fixed this issue by modifying the following :

- the loop-up tables (LUT) have been recalculated to cover a larger spectral range (960-1105 cm$^{-1}$ versus 960–1075 cm$^{-1}$), with correcting numerical implementation in FORLI v20151001, especially with regard to the LUTs at higher altitude.
- the HITRAN spectroscopy database was updated from HITRAN 2012 to HITRAN 2004

But these changes are not fully efficient at high latitudes, and as the reviewer noticed, the exact cause of the discrepancy left is still under investigation. To answer the comments in more details, we added the following arguments to the revised manuscript:

"*Despite the global improvement of ~5 % with the new IASI TOC product with respect to the previous IASI TOC product (v20140922), large discrepancies are still observed at high latitudes and are partly explained by:*

i)     *the low spectral signal to noise ratio due to very low surface temperature in this region leading to limited information content in the IASI observations in these regions;*

ii)    *a misrepresentation of the wavenumber-dependent surface emissivity, which is a critical input parameter to describe the surface, especially above continental surfaces (Hurtmans et al., 2012). FORLI uses the emissivity climatology built by Zhou et al. (2011) providing weekly emissivity values on a 0.5°x0.5° latitude/longitude grid for all 8461 IASI spectral channels. However, Zhou et al. climatology can have missing values. In such cases, the MODIS climatology built by Wan (2006), which provides values for only 12 channels in the IASI spectral range is used instead. Furthermore, in case of no correspondence between the IASI pixel and either climatologies, the reference emissivity used for the Zhou climatology (Zhou et al., 2011) is used, which can significantly impact the retrievals, in particular in arid or semi-arid regions where variations in emissivity are large both on spectral and spatial scales (Capelle et al., 2012) but also in ice region since the reference emissivity does not necessarily reflect the actual snow or sea ice coverage;*

iii)   *the temperature profiles used in FORLI-O$_3$ that are less reliable at high latitudes and over elevated terrain (August et al., 2012). As shown in Boynard et al. (2009), the errors introduced by the uncertainties of 2 K on the temperature profile can reach up to 10 % of total error on the retrieved vertical profile, with the error due to the temperature uncertainty on the TOCs being much lower.*

> *Errors on thermal contrast can also have an impact on the retrievals.*
>
> iv)    *the errors associated with TOC retrievals in the UV-vis spectral range increasing at high solar zenith angles in these regions, mostly because of the larger sensitivity of the retrieval to the a priori $O_3$ profile shape (Lerot et al., 2014).*
>
> *In the section below, a detailed analysis of the larger bias found in the Antarctic region is undertaken for individual ground-based Brewer and Dobson station to try to understand the larger bias (see next section).*"

**10. P8 L1 Is there some explanation what the physical basis is for the rejection of data over deserts and Antarctica?**

Data characterized by a C-shape profile, which is not realistic, are generally located over desert and Antarctica. A possible explanation of this issue is a misrepresentation of emissivity above sand and ice surfaces. More explanations on the impact of misrepresented emissivity on ozone retrievals are given in the previous answer and were added to the revised manuscript.

**11. P11 L18 Its not completely clear are all comparisons of the upper most partial column to 10hPa despite the statement to the contrary on L17 (25-3hPa)?**

As ozonesonde profiles sonde generally burst around 30-35 km, the middle stratosphere upper was limited to 10 hPa. We made it clearer in the revised manuscript (new comparison methodology section) as follows:

*3. Comparison methodology*

"*The IASI-A and IASI-B $O_3$ products are assessed in terms of TOCs and partial ozone columns. The validation exercise is performed using the same partial columns as those used in Wespes et al. (2016) since these columns contain around one piece of information, have maximum sensitivity approximately in the middle of each of the layers, and reproduce the well-known cycles related to chemical and dynamical processes characterizing these layers: surface-300 hPa (TROPO), 300-150 hPa (UTLS), 150-25 hPa (LMS) and 25-3 hPa (MS). On average, these pressure columns correspond to the following altitude columns: surface-8 km, 8-15 km, 15-22 km 22-40 km, respectively. Note, however, that for the comparison between IASI and ozonesonde data, the MS is limited to the column 25-10 hPa as sonde generally burst around 30-35 km (see Section 3.2 below). For the assessment of IASI vertical profiles, we refer to Keppens et al. (2018, this issue).*"

**12. P11 L22 What is the source of the extremely low O3 in the UTL? Is it low in the sonde, IASI or both? How much data is lost?**

The extremely low $O_3$ in the UTLS (<1 DU) concerns the sonde smoothed with the IASI averaging kernels. Up to 8 % of the data in the tropical UTLS are lost. We added this information to the revised manuscript as follows:

"*To avoid unrealistic statistics skewed by extremely unrealistic low values in the UTLS $O_3$ columns found in the smoothed ozonesonde data, we filter out extreme outliers exceeding 200 % relative differences with IASI (which can be up to ~8 % of the data in the tropical UTLS).*"

**13. P12 L8 It is very reasonable to follow that the high variability in the UTLS give large SD but less so to see what the a priori has to contribute given DOFS 1.**

We changed to sentence to:

" *The standard deviation is maximum in the UTLS in all latitude bands (compared to the other partial columns) due to strong $O_3$ variability and large total retrieval error as shown in Wespes et al. (2016).*"

**14. P12 L14 Its not clear what information is missing? The comparison is to 10hPa for both instruments – is that correct?**

Yes that is correct. We made this point cleared in the revised manuscript as follows:

*"The correlation coefficient ranges between 0.6 (tropics and high latitudes) and 0.8 (mid-latitudes) for the LMS column while they are much lower for the MS column, which is explained by the low DOFS values ranging between 0.4 and 0.6 as indicated on the scatter plots. Note that the DOFS for the MS columns are lower than those calculated in Fig. 13 because they do not correspond to the full MS column calculated from IASI (25-3hPa i.e. ~25-40km) but to the MS columns truncated to match the maximum altitude (30-35km) of the sonde measurements."*

15. P12 L29 Please clarify are these 40 pairs globally?
We rephrased to sentence to:
*"The long-term stability of IASI-A partial $O_3$ column vs ozonesonde measurements is assessed in Figure 16, which presents the monthly relative differences between IASI-A and ozonesonde for the TROPO, UTLS, LMS and MS $O_3$ partial columns for a total of 18 ozonesonde stations in the NH that cover eight years or longer (over 2008 – 2017). With more than 30 IASI-sonde pairs per month, the NH presents sufficient collocated data to assess a good statistical drift analysis on the contrary to the SH (only 8 ozonesonde stations)."*

16. P13 L21 Please provide a reference for the stability of the IASI radiances.
We removed this sentence.

17. P13 L31 use of the adjective generally is not helpful.
We removed the word 'generally'.

Technical:

1. P1 L19 what does ir "generally consistent" versus just "consistent"
We have changed 'generally consistent' to 'consistent'

2. P1 L23 "retrieved ones" would better be "retrieved TOC's"
As highlighted by Referee #2, Brewer/Dobson data are not retrieved products. We removed the word "retrieved" and changed "ones" to "TOCs".

3. P1 L24 "on the instruments" would better be "on the compared instruments"
We have made the change.

4. P1 L30 (up to 20% for the 150-25 hPa column) should be (up to 20% for the 150-25 hPa partial column)
Following a previous comment of the referee, this sentence has been rephrased.

5. P1 L30 "within" might better be "less then"
We have changed 'within' to 'less than'.

6. P2 L 4 "amount" should be "amounts"
This has been corrected.

7. P9 L2 'latitude belt' might be better worded 'latitude band'
This has been corrected.

8. P3 L11 "overestimates the ultraviolet (UV) Total Ozone Column (TOC)" Do you really

mean to differentiate the uv TOC from some other spectral region?

Only comparisons between IASI and UV-vis Total Ozone Column have been performed so far that is why we only refer to bias between IASI and UV-vis data. However this does not mean we differentiate the UV TOC from other spectral region such as TIR. We added this sentence:

"*It is worth mentioning that Boynard et al. (2016) did not perform any comparison with measurements in other spectral ranges than UV.*"

9. P5 L19 'posteriori' should be 'after'

We have made the change as follows: '*after September 2015*'

10. P5 L 31 remove second 'between April. . .'

We have made the change.

11. P6 L1 'proved' might better be worded 'shown'. Fruthermore is there a reference
for this statement/conclusion?

This has been corrected. We added references for this statement.

12. P6 L13 'being preferentially be used' might better be worded 'are recommended'.

This sentence has been changed to:

"*Even if for the period April – September 2015, IASI-B O$_3$ products are better recommended for a high quality use, it is worth noting that the IASI-A instrumental issue only affects the TOC by 0.4 % and the tropospheric ozone by 10 %, which are much lower than the TOC and tropospheric total retrieval errors estimated to 2 % and 20 % on average, respectively, justifying the potential use of the IASI-A data over that period if it is required. In the validation exercise presented in the next section, the period April-September 2015 is included.*"

13. P12 L20 From plot 14 I read +1.5 & – 3.5% difference (not +- 3.5) and SD of maximum 14.6% (not 14)

We changed the sentence to:

"*The comparison is good for all latitudes, with IASI-A O$_3$ products underestimating the TROPO O$_3$ abundance in the mid-latitudes and tropics by ~1.6-3.5 DU (7.1-14.3 %) and overestimating the TROPO O$_3$ abundance in the high latitudes by ~1.5 DU (4–7 %).*"

14. P12 L 23 Dobson units (sp)

This has been corrected.

15. P12 L28 . . .O3 partial columns. . .

The word "partial' has been added.

16. P13 L20 . . .proven to be very. . .

This sentence has been removed.

17. P14 L7 'better from October' might be better worded 'better after October'

Since the change in the IASI-A viewing angle was corrected in September, we have changed 'better from October' to 'better after September'

18. P33 caption, use of the term 'sub' column should be removed (in all cases) and partial be used for consistency

We have changed 'sub-column' to 'partial column'

---

## Author Comment (AC2) · 1 Jun 2018

**Response to Referee #2**

**The authors would like to thank the referee for her/his constructive and detailed comments, which have helped us clarifying several points and improving the manuscript. Below are our responses to the comments brought up by the referee. Referee's comments and our replies are marked in blue and in black, respectively. In italic are the changes made in the manuscript.**

General comments
The manuscript titled "Validation of the IASI FORLI/Eumetsat ozone products using satellite (GOME-2), ground-based (Brewer-Dobson, SAOZ) and ozonesonde measurements" give a thorough validation study of 9 years of IASI ozone measurement. The manuscript is well written, clear and easy to read.

However, it is not easy to understand whether this paper (Boynard et al., 2017) presents novel concepts, ideas, tools or data, especially when we compare Boynard et al. (2017) to Boynard et al. (2016). Many conclusions in Boynard et al. (2016) and in Boynard et al. (2017) are similar.

Were the authors expecting different results between IASI v20140922 and v20151001? According to Boynard et al. (2016), the improvement of IASI retrieval was already found to be mainly located in the middle stratosphere. How much could the bias assessment change with 2 more years of data? 9 years of data allow the authors to address the long-term stability of IASI. This is the most interesting and the newest part of the study. Unfortunately the significant drift in the troposphere is barely explained and addressed.

Boynard et al. (2016) validated IASI data processed with the previous version of FORLI (v20140922) on a global scale over 7 years of data (2008-2014), and showed a constant bias in the TOCs between IASI and other datasets, of the order of 4-6%, depending on the datasets. Some draft corrections were implemented in FORLI, and preliminary comparisons limited to only 12 days, showed an improvement of 2-4% compared to the former FORLI version, which is mainly related to an improvement in the middle stratosphere (Boynard et al., 2016). Since the data retrieved using FORLI-$O_3$ v20151001 will become the official Eumetsat product in 2018, it is required to validate the full available IASI dataset and not only 12 days taken randomly, which is not representative of all atmospheric conditions. Furthermore, in the framework of European projects such as CCI and C3S projects focusing on building consolidated climate-relevant ozone data sets as essential climate variables (ECVs), it is necessary to validate satellite data for a long-time period, which is one of the goal of the current manuscript.

For clarity purpose, in the revised manuscript, we better explain the goal of this manuscript in the introduction. We quantify in detail the improvements of the new version of FORLI v20151001 in comparison with the previous one v20140922. The new version allows to remove the systematic bias that was identified with the former version. Local discrepancies identified earlier persist e.g. at high latitudes, and over mountain region and desert. that couldn't be fixed with the current version but quality flags allow to filter them if needed.

Here are the changes made in the revised manuscript:

Introduction:
"This $O_3$ retrieval algorithm (FORLI-$O_3$ v20151001) is currently being implemented into the Eumetsat processing facility under the auspices of the Ozone and Atmospheric Composition Monitoring Satellite Application Facility (AC SAF) project in order to operationally distribute Level-2 IASI $O_3$ profiles to users through the EumetCast system in 2018. *IASI Level-2 and Level-3 $O_3$ products processed with FORLI v20151001 are part of the European*

*Space Agency O₃ Climate Change Initiative (Ozone_cci, www.esa-ozone-cci.org) and the European Centre for Medium-Range Weather Forecasts (ECMWF) Copernicus Climate Change (C3S) projects, respectively, which focus on building consolidated climate-relevant ozone data sets as essential climate variables (ECVs). Therefore, validating the latest version of the IASI O₃ products over a long-time period and assessing their stability are necessary for decadal trend studies, model simulation evaluation and data assimilation applications. This is one of the main motivations of the present work. The goals of the Ozone_cci project are described in Garane et al. (2018) and its requirements in term of satellite product stability, which is defined to 1 – 3% / decade based on the requirements formulated by the Global Climate Observing System (GCOS) and the Climate Modeling User Group (CMUG) climate modelling community for ozone is detailed in Van Weele et al. (2016)."*

GOME-2 comparison section:
"*Globally, IASI-A (IASI-B) TOC product are slightly higher than GOME-2A TOC product, with a global mean bias of 0.3±0.8 % (0.4±0.8 %). It is worth noting that the previous IASI TOC product (v20140922) was in disagreement by more than 5 % (Boynard et al., 2016). The global mean bias is now within total errors of GOME-2 estimated to 3-7 % (Valks et al., 2017) and IASI (e.g. Boynard et al., 2009), which demonstrates the good consistency between IASI and GOME-2 TOC products.*"

Brewer/Dobson comparison section:
"*Nevertheless the overall comparison with Dobson and Brewer TOCs shows that IASI new TOC product is improved by 4% in comparison with the previous IASI TOC product (v20140922; see Boynard et al. (2016)) and is within IASI and GB TOC total error bars.*"

SAOZ comparison section:
"*The results are consistent with those found for the comparison with GOME-2A along with Brewer and Dobson measurements (see Sections 5.1 and 5.2, respectively). An improvement of 3-4 % is found when compared to the previous IASI product (v20140922).*"

Ozonesonde comparison section:
"*In comparison with the previous IASI partial ozone column products reported in Boynard et al. (2016), the new IASI ozone product is significantly improved in the MS by 8-12 % for the mid latitudes and tropics. The improvement is less significant for the LMS except in Antarctic where an improvement of 6 % is found. As for the TROPO and UTLS columns, no or slight improvement (<2 %) is found, and the agreement between IASI and sonde data is even worse compared to the previous IASI ozone product, especially for the southern tropical TROPO column (by 7 %) and the UTLS column (by 10-18 %).*"

As highlighted by the referee a novel part of this manuscript is the study of the long-term stability of IASI, which is essential for long-term analysis of stratospheric and tropospheric ozone, such a decadal trend studies, model simulation evaluation and data assimilation applications. A significant and surprising drift has been found in the IASI tropospheric dataset for the period 2008-2016. When considering the period 2011 – 2016, the drift value for the TROPO column decrease and become statistically insignificant. However, since this difference in the drift values might be due only to the too short periods considered here associated with the high variability in TROPO O₃ differences, a few more years are needed to confirm the observed negative drifts and evaluate it on the longer term.

Another new in this manuscript (compared to Boynard et al. 2016) is the assessment of the significant difference in L1 data between IASI-A and IASI-B between April and September 2015, which was not possible to show in

Boynard et al. (2016) since the validation period extended from 2008 through 2014.

Furthermore, Boynard et al. (2017) didn't address open questions already mentioned in the conclusions of Boynard et al. (2016), such as the large bias found in the UTLS, still not fully understood.
We now address this question in the new Section on IASI/FTIR comparison as follows:
"*It should be noted that IASI is positively biased in the UTLS region, as reported in previous studies (e.g. Dufour et al., 2012; Gazeaux et al., 2013). Although Dufour et al. (2012) attempted to give some explanations for this particular feature, the exact reason for this overestimation is still not clear. One reason could be the use of inadequate a priori information. Note that FORLI uses only one single a priori profile (Hurtmans et al., 2012) that is the global mean profile of the McPeters/Labow/Logan climatology (McPeters et al., 2007). As shown by Bak et al. (2013), using tropopause-based ozone profile climatology can significantly improve the a priori. However, using dynamical a priori makes the comparison on a global scale less straightforward since a different a priori profile would be used at each IASI pixel.*"

For these reasons, I would suggest major corrections before the current manuscript can be published in AMT.

Specific comment:

- Section 3: Intercomparison between IASI-A and IASI-B ozone products

Line 18 p. 5: Change "the figure" to "Figure 2"
We changed "the figure" by "*Fig. 2*" according to AMT guidelines.

Line 2 p. 6: Change "then" to "October 2015"
We have made the change.

Line 14 p. 6: April-October 2015 shouldn't be included in the combined IASI-A/IASI-B product (as explained in Line 10), because of instrumental issues on IASI-A. Should this time-period be excluded from any time-series studies with IASI-A?

It depends on the interest of the user. If the user wants to analyze the total ozone column from IASI-A, he has to be aware of that issue but it is worth noting that the total column is only affected by 0.4%, which is well below the ozone column retrieval error, estimated to ~2% globally**.** Furthermore, Wespes et al. (2018) who performed tropospheric ozone trend study did not exclude this short 6-month period, which is relatively short over the 10 years of IASI-A data and therefore is not supposed to affect the calculation of the trend in tropospheric ozone. The instrumental issues on IASI-A affect the tropospheric ozone up to 10%, however again this is lower than error bars for tropospheric ozone**,** estimated to 20% globally.

We added the following paragraphs to the revised manuscript:

Section 4:
"The excellent agreement between the current IASI-A and IASI-B TOC and TROPO $O_3$ columns (April – September 2015 excluded) allows the combined use of IASI-A and IASI-B instruments to provide homogeneous total and tropospheric ozone data with full daily global coverage measurements. *Even if for the period April – September 2015, IASI-B $O_3$ products are better recommended for a high quality use, it is worth noting that the*

*IASI-A instrumental issue only affects the TOC by 0.4% and the tropospheric ozone by 10%, which are much lower than the TOC and tropospheric retrieval errors estimated to 2 % and 15 % on average, respectively, justifying the potential use of the IASI-A data over that period if it is required. In the validation exercise presented in the next section, the period April-September 2015 is included.*"

Summary:
"*However, it is worth noting that the impact of IASI-A instrumental issue is within the TOC and TROPO O₃ column total error bars. In case of using IASI-A data only, the user is free to include or exclude the period April – October 2015 depending on the interest of the study.*"

- Section 4: Validation of IASI-A and IASI-B total ozone columns
I would suggest to move all the validation method in one method section. The method section could then include: (1) the formulae of differences calculation, (2) The method of co-location between IASI and reference observations, (3) the characteristics of the data used for the comparison. This change would help the authors to shorten several sub-sections.

We  followed the referee's suggestion. The revised manuscript includes the following new sections:
- **2. IASI measurements and independent datasets used for the validation**: which describes the IASI $O_3$ retrievals as well as the independent measurements used for the validation
- **3. Comparison methodology**: this section includes the formulae of difference calculation as well as the different comparison methods (including the co-location criteria) used in the present work.

Line 25-27 p. 7: This statement is almost word to word the same as in Boynard et al. (2016). Don't the author think that "Further investigation would be needed to understand the reasons of these larger differences at high latitude" should be addressed in the current study?

As explained before ,this manuscript validates a different product from Boynard et al. (2016) study.  A detailed analysis was undertaken for individual ground-based station located in Antarctic and desert regions characterized by larger biases. However this analysis examining the dependency of the relative differences on the parameters available in FORLI outputs (viewing zenith angle, Root-Mean_Square Error, TOC error, distance from the ground station and DOFS) was not concluding. Actually, the stations located in desert show a confusing behavior with positive (Tamanrasset) and negative (Aswan and Springkok) biases of 7 -8  % and 4-5%, respectively. As for Antarctica, four stations were examined: the bias is extremely high for Amundsen-Scott located at 90° S and 3 km altitude (20%) and less, but still positive, for the other three ice-covered stations Haley-Bay, Syowa, Arrival-Heights (1.2 -3.8 %). The comparison of GOME-2A with ground-based TOCs at Amundsen-Scott shows a bias of 1-2 % indicating there is no issue in the ground-based measurements. Furthermore the scatter plot for that particular station (compared to either Dobson or Brewer) shows that IASI-A has a much higher variability than the ground-based measured TOC values. A future work will be initiated with different groups involved in validation studies in order to further examine the origin of the bias for the location of interest (Antarctica, mountain and desert region), taking into account the measurement dependence on surface emissivity as well as other parameters.

We rephrased this part as follows:
"*Despite the global improvement of ~5% with the new IASI TOC product with respect to the previous IASI TOC product (v20140922), large discrepancies are still observed at high latitudes and are partly explained by:*
- *i)    the low spectral signal to noise ratio due to very low surface temperature in this region leading to limited information content in the IASI observations in these regions;*
- *ii)   a misrepresentation of the wavenumber-dependent surface emissivity, which is a critical input parameter to describe the surface, especially above continental surfaces (Hurtmans et al., 2012).*

*FORLI uses the emissivity climatology built by Zhou et al. (2011) providing weekly emissivity values on a 0.5°x0.5° latitude/longitude grid for all 8461 IASI spectral channels. However, Zhou et al. climatology can have missing values. In such cases, the MODIS climatology built by Wan (2006), which provides values for only 12 channels in the IASI spectral range is used instead. Furthermore, in case of no correspondence between the IASI pixel and either climatologies, the reference emissivity used for the Zhou climatology (Zhou et al., 2011) is used, which can significantly impact the retrievals, in particular in arid or semi-arid regions where variations in emissivity are large both on spectral and spatial scales (Capelle et al., 2012) but also in ice region since the reference emissivity does not necessarily reflect the actual snow or sea ice coverage;*

iii) *the temperature profiles used in FORLI-O$_3$ that are less reliable at high latitudes and over elevated terrain (August et al., 2012). As shown in Boynard et al. (2009), the errors introduced by the uncertainties of 2 K on the temperature profile can reach up to 10% of total error on the retrieved vertical profile, with the error due to the temperature uncertainty on the TOCs being much lower. Errors on thermal contrast can also have an impact on the retrievals.*

iv) *the errors associated with TOC retrievals in the UV-vis spectral range increasing at high solar zenith angles in these regions, mostly because of the larger sensitivity of the retrieval to the a priori O$_3$ profile shape (Lerot et al., 2014).*

*In the section below, a detailed analysis of the larger bias found in the Antarctic region is undertaken for individual ground-based Brewer and Dobson station to try to understand the larger bias (see next section).*"

Futhermore, we added a description of the detailed analysis undergone for individual ground-based station in the Brewer/Dobson section:

"*To further examine the large discrepancies mentioned above, we have analyzed in more details the results obtained for individual stations located in Antarctic and desert regions. The stations located near desert areas show an diverging behavior with positive (Tamanrasset, Algeria) and negative (Aswan, Egypt and Springbok, South Africa) biases of +7 to +8% and -5 to -4%, respectively. Over Antarctica, four stations were examined: the bias was found to be extremely high for Amundsen-Scott located at 90° S and 3 km altitude (20%) and less, but still positive, for the other three stations Haley-Bay, Syowa, Arrival-Heights (1.2 – 3.8 %) located on the Antarctic Ice Sheet. The comparison of GOME-2A with ground-based TOCs at Amundsen-Scott shows a very small bias of 1-2%, indicating there is no obvious issue with the ground-based measurements. Furthermore, the scatter plot for that particular station (compared to either Dobson or Brewer; plot not shown) shows that IASI-A has a much higher variability than the GB TOC values. This issue has still to be further explored by investigating, for instance, the impact of potential surface emissivity discrepancies on the retrievals over some regions of Antarctica and deserts. Additional quality filters, e.g. on ice surface emissivity issues, could also be considered.*"

Line 33 p. 7: Would it be possible to quantify the "better agreement"?
The agreement is better by ~5%. We added the following sentence to the revised manuscript as follows:
"*It is worth noting that the previous IASI TOC product (v20140922) was in disagreement by more than 5 % (Boynard et al. , 2016). The global mean bias is now within total errors of GOME-2 estimated to 3-7 % (Valks et al., 2017) and IASI (e.g. Boynard et al., 2009), which demonstrates the good consistency between IASI and GOME-2 TOC products.*"

Following a comment of Referee #1, we better quantify the improvements of the new version of FORLI in comparison with the previous version in the revised manuscript. We also highlight the fact that the improvement is rather constant over the globe and therefore issues are still persisting over some regions such as high latitudes, mountain and desert regions.

Lines 3-11 p. 8: All this paragraph is already stated in Boynard et al. (2016). It could be either removed or shortened.
As suggested by the referee, we removed this paragraph and Figure 8.

Line 10 p. 8: Would it be possible to quantify the "magnitude"?
The paragraph has been removed as suggested by the previous referee's comment.

Line 28 p. 8: Could you explain why the number of stations would influence the dependency on the latitude of the differences between IASI and GB measurements?
As discussed in the manuscript, the mean difference for each 10° latitude bin is calculated using all percentage daily differences between GB and satellite measurements located in the respective bin. In the Southern Hemisphere there are bins that include only a few stations, so if one or two of them is not in great agreement with the satellite measurements, the mean value of the whole bin appears to deviate strongly from the 0% line. This is not the case for the Northern Hemisphere, where many stations contribute to each latitude bin and the incompatibility of one or two of them would not have a strong influence on the latitude mean.

We changed this sentence to:
"*As shown by the IASI-to-Dobson comparison (left panel), the dependency on latitude is less visible for the NH due to the high number of collocations which renders the latitudinal means more representative compared to the SH.*"

Line 29 p. 8: The differences between IASI-A and Dobson seem to reach 3.5% in NH, while the authors report [0-2.5%]
We made the correction and the sentence has been changed as follows:
"*The comparisons with Dobson measurements show differences between 0 and 2.5 % for the entire NH (except in the 70-80°N belt where difference reaches 3.5 % for IASI-A) and for latitudes ranging between 0° and 40° S. *"

Line 30 p. 8: "Lower than 40°S" would mean somewhere between 0 and 40°S. Do you mean between 40°S and 60°S? Please clarify.
We have changed "lower" by "*Southwards of*".

Line 1 p. 9: It is worth to notice there is no Brewer measurements in SH.
Actually there are Brewer stations in the SH, but they are not evenly distributed. We changed the sentence to:
"*A similar picture for the NH is observed for the comparison with Brewer measurements. Note that there are a few Brewer stations in the SH, but they are not evenly distributed (all of them are located on the Antarctic) so their measurements are not used.*"

Line 2 p. 9: Change "belt" to "region'.
Done.

Lines 5-6 p. 9: Could you explicitly mention the 1-3% requirement from the Ozone_cci project instead of " within ±3%":
We changed "within 3%" by "*among 1 and 3%*".

Line 11 p. 9: Change "small" to " < 3% ".
Done.

 In the (new) method section I would suggest to explain the ozone-cci project and their requirement in term of satellite products stability. Could you explain how the 1-3% requirement has been decided? According to the 1-3% requirement, "IASI-A TOC products are reliable for trend studies". Does it mean no drift adjustment at all is required? And does it mean that the drift, even small, is not taken into account in the ozone trends uncertainties? Could you please explain?

The Ozone_cci project is fully described in Garane et al. (2018, this issue) and their requirement in term of satellite product stability are detailed in Van Weele et al. (2016). We now refer to both references in the introduction of the revised manuscript as follows:

"*This $O_3$ retrieval algorithm (FORLI-$O_3$ v20151001) is currently being implemented into the Eumetsat processing facility under the auspices of the Ozone and Atmospheric Composition Monitoring Satellite Application Facility (AC SAF) project in order to operationally distribute Level-2 IASI $O_3$ profiles to users through the EumetCast system in 2018. It is therefore essential to validate the full available IASI dataset and not only 12 days taken randomly, which is not representative of all atmospheric conditions as it was performed in Boynard et al. (2016). Furthermore, IASI Level-2 and Level-3 $O_3$ products processed with FORLI v20151001 are part of the European Space Agency $O_3$ Climate Change Initiative (Ozone_cci, www.esa-ozone-cci.org) and the ECMWF Copernicus Climate Change (C3S), respectively, which focus on building consolidated climate-relevant ozone data sets as essential climate variables (ECVs). Therefore, validating the latest version of the IASI $O_3$ products over a long-time period and assess their stability are necessary before using the data for scientific applications such as assimilation or trend studies. This is one of the main motivations of the present work. The goals of the Ozone_cci project are described in Garane et al. (2018) and its requirements in term of satellite product stability, which is defined to $1 - 3\%$ / decade based on the requirements formulated by the Global Climate Observing System (GCOS) and the Climate Modeling User Group (CMUG) climate modelling community for ozone is detailed in Van Weele et al. (2016).*"

As for the comment on drift adjustment, the Ozone_cci project is responsible for producing long term homogenized ozone data sets, which can (but might not) have a maximum of 1-3% drift. It is not the responsibility of the Ozone_cci data providers to correct for any drift within or higher than this limit, if this exists.

 Which criteria is used to qualify differences "within 1.1%" as "very good agreement"?
We removed the word "very".

 This paragraph is not clear. It is hard to understand what would explain differences in the seasonal variability between Dobson and Brewer. What does 0.6% represent?
The 0.6% difference represents the expected difference between TOC measurements from Brewer and Dobson spectrometers.

The paragraph was rephrased as follows:
"*Fig. 10 shows a good agreement between IASI-A and GB measurements (mean differences within 1.1%), with an obvious seasonal variability in the differences: the smallest differences appear in summer and the largest differences in winter. In the Dobson comparison the seasonal variability is more evident, which is explained by the fact the TOC measurements from Dobson spectrometers depend strongly on the stratospheric effective temperature (Koukouli et al., 2016). We can also see a similar but less pronounced seasonality effect in the Brewer comparison. According to Garane et al. (2018) and references therein, even though Dobson and Brewer spectrometers follow almost the same principles of operation, TOC measurements from the two types of instruments show differences in the range of ±0.6% due to the use of different wavelengths in their respective TOC algorithms and the different*

*temperature dependence for the ozone absorption coefficients. However it is worth noting that these differences between Brewer and Dobson TOCs are lower than their total uncertainty (~1%). The mean difference for the NH is lower than 1.1 % for both Dobson and Brewer comparisons to the IASI observations."*

- Section 5: Validation of IASI-A and IASI-B partial ozone column products
As mention for Section 4, I would suggest to move the comparison method in one method section.
We followed the referee's suggestion.

Line 16 p.10: Could you report the numbers of the "small or non-significant negative decadal trends"?
We changed the sentence to:
*"The IASI-A and SAOZ TOC relative differences show small or insignificant negative decadal trends ranging between -0.02±0.65% (OHP) and -2.06±0.66% (Reunion), except for Bauru station. The good quality of the IASI-A TOC temporal stability satisfies well the $1 - 3\%$ decade$^{-1}$ Ozone_cci requirements for the long-term stability for total ozone measurements (Van Weele et al., 2016), which shows again that the current IASI-A TOC products are homogeneous and reliable for trend studies."*

Lines 16-17 p.10: Could you refer again to the 1-3% requirement with the reference of the Ozone_cci project?
The sentence has been changed as follows:
*"The good quality of the IASI-A TOC temporal stability satisfies well the $1 - 3\%$ decade$^{-1}$ Ozone_cci requirements for the long-term stability for total ozone measurements (Van Weele et al., 2016), which shows again that the current IASI-A TOC products are homogeneous and reliable for trend studies."*

Line 27 p. 10: "[…] their uncertainties are lower than other types of ozonesondes […]" Could you quantify?
The sentence has been changed as follows:
*"Their accuracy is generally good (±3-5%) and their uncertainties are of about 10% throughout most of the profile below 28 km (Deshler et al., 2008; Smit et al., 2007), while other types of ozonesondes have somewhat poorer accuracy (5-10%), (e.g. Hassler et al., 2014; Liu et al., 2013)."*

Lines 6-12 p. 11: The common method to compare satellite data with ozonesondes is to degrade the high vertically resolved ozonesondes by applying the AKs and a priori ozone profiles used to retrieve satellite ozone products. In Huang et al. (2017), they use the high vertically resolved ozonesondes (without degrading the vertical resolution) in the regions and altitudes when the satellite has low retrieval sensitivity. Could you comment on this? Is such analysis could be done in your study?

Huang, G., Liu, X., Chance, K., Yang, K., Bhartia, P. K., Cai, Z., Allaart, M., Ancellet, G., Calpini, B., Coetzee, G. J. R., Cuevas-Agulló, E., Cupeiro, M., De Backer, H., Dubey, M. K., Fuelberg, H. E., Fujiwara, M., Godin-Beekmann, S., Hall, T. J., Johnson, B., Joseph, E., Kivi, R., Kois, B., Komala, N., König-Langlo, G., Laneve, G., Leblanc, T., Marchand, M., Minschwaner, K. R., Morris, G., Newchurch, M. J., Ogino, S. Y., Ohkawara, N., Piters, A. J. M., Posny, F., Querel, R., Scheele, R., Schmidlin, F. J., Schnell, R. C., Schrems, O., Selkirk, H., Shiotani, M., Skrivánková, P., Stübi, R., Taha, G., Tarasick, D. W., Thompson, A. M., Thouret, V., Tully, M. B., Van Malderen, R.,
Vömel, H., von der Gathen, P., Witte, J. C., and Yela, M.: Validation of 10-year SAOOMI Ozone Profile (PROFOZ) product using ozonesonde observations, Atmos. Meas. Tech., 10, 2455-2475, 10.5194/amt-10-2455-2017, 2017.

In the framework of a validation study, it is  appropriated and recommended to use the averaging kernels to take into account the differences in the sensitivity profiles. Indeed, comparing raw products as performed by Huang et al. (2017) is interesting, however IASI profiles have much less vertical information than ozonesonde profiles and thus a direct comparison is not recommended. Furthermore, a direct comparison between raw products only gives an indication on the sensitivity but does not affirm there is a lack of sensitivity. It is clear that if there is no sensitivity the smoothed profile will reproduce the *a priori* profile and therefore this is not interesting to analyze these cases for validation purpose.

In some cases, it is clear that the lack of information implies that the smoothed ozonesonde approaches the *a priori*, which does not offer any interest for validation purpose. However, as shown in the figure below illustrating the comparison between IASI-A against smoothed (blue) and raw (red) ozonesonde data, the smoothed data does not reproduce the *a priori* in case of low DOFS, which means that a minimum of information is brought by IASI. We chose to not add this figure in the paper given that first IASI and sonde data have significantly different sensitivity profiles and, second, IASI always has a minimum of sensitivity that justifies the use of AK for the comparison. To indicate if there is sensitivity or not, comparing raw and smoothed data does not allow to indicate if there is sensitivity or not but DOFS gives this information. The DOFS are indicated in Figure 14 of the manuscript so that the reader can know the region/altitude characterized by low sensitivity. We recommend to give less attention and interest to comparison results when IASI presents no sensitivity.

[Figure]

Figure: Scatter plots of IASI-A against smoothed (red) and raw (blue) sonde O₃ partial columns for six latitude bands for the period January 2008 – July 2017. Comparison statistics including the linear regression, the mean differences and standard deviation in both Dobson units and percent, the number of collocations and the mean DOFS for each partial column are shown on each panel.

Lines 18 and 31 p. 12: The selection of the ozonesondes stations are confusing. Why don't you use all the ozonesondes stations that meet the criteria needed for the comparison such as long-term time series, statistics of

the data, etc: : :?

Figure 14 just illustrated a sample of time series of IASI and smoothed ozonesonde data for six stations representative of different latitude bands and with data available over the period 2008 – 2017.

In order the avoid spurious effects due incomplete annual cycle and/or characterized with too short temporal coverage, only time series of eight years or longer are used for the assessment of IASI-A temporal stability. As shown in the figure below, several ozonesonde stations are characterized by too short temporal coverage or incomplete annual cycle, and including these stations in the drift calculation will bias the results.

[Figure]

Figure: (left panels) Time series of daily IASI-A (in red) and smoothed ozonesonde (in blue) TROPO $O_3$ columns for six stations characterized by incomplete annual cycle or too short temporal coverage between 2008 – 2017; (right panels) Associated relative differences (in percent), including the mean differences and 1-sigma standard deviation.

We rephrased Line 30-31 p.12 to:

*"The long-term stability of IASI-A partial $O_3$ column vs ozonesonde measurements is assessed in Figure 16, which presents the monthly relative differences between IASI-A and ozonesonde for the TROPO, UTLS, LMS and MS $O_3$ partial columns for a total of 18 ozonesonde stations in the NH that cover eight years or longer (over 2008 – 2017). With more than 30 IASI-sonde pairs per month, the NH presents sufficient collocated data to assess a good*

*statistical drift analysis  on the contrary to the SH (only 8 ozonesonde stations).*"

 This part of the discussion is one of the most interesting but it is too short. Would it be possible to address at least one of the speculative explanation for such a drift?
The ozonesonde-to-IASI comparison shows that the drift values calculated for two different periods (2008 – 2016 and 2011 – 2016) differ. Since the difference in the drift values between the two periods might be due only to the too short periods considered here (9 years or less) associated with the high variability in the TROPO $O_3$ differences, a few more years are needed to confirm the observed negative drifts and evaluate it on the longer term. Another possible but speculative explanation for this drift in the TROPO is the changes in the IASI Eumetsat L2 dataset version over the IASI time period. However we do not prefer to include this explanation in the manuscript because it is very speculative and cannot be confirmed without using homogeneous L2 dataset, which is planned for the future (it takes 2 years to reprocess the whole IASI data record).

These lines have been changes to:

"*Note that for the TROPO column, the drift calculated for each individual station ranges between -16 % decade$^{-1}$ and -5 % decade$^{-1}$, which is the same order of magnitude of those found in the IASI-A to FTIR TROPO comparison. If we limit the time period to 2011 –2016, no statistically significant drift is found anymore for the TROPO and MS (P value >0.47). However, since this difference in the drift values might be due only to the too short time periods considered here associated with the high variability in the TROPO $O_3$ differences, a few more years are needed to confirm the observed negative drifts and evaluate it on the longer term.*"

- Summary
 Would you suggest to remove the data between April and September 2015 (October 2015 in the main text) for trends studies? If so, could you mention it? Would it be possible to apply any corrections factor on the data for this time-period?
It depends on the interest of the user. If the user wants to analyze the total ozone column from IASI-A, he has to be aware of that issue but it is worth noting that the total column is only affected by 0.4%, which is well below total ozone column total error, estimated to 2% globally. Furthermore, Wespes et al. (2018) who performed tropospheric ozone trend study did not exclude this short 6-month period, which is relatively short over the 10 years of IASI-A data and therefore is not supposed to affect the calculation of the trend in tropospheric ozone. The instrumental issues on IASI-A affect the tropospheric ozone up to 10%, however again this is lower than error bars for tropospheric ozone, estimated to 20% globally.

We added the following sentence to the revised manuscript:
"*In case of using IASI-A data only, the user is free to include or exclude the period April – October 2015 depending on the interest of the study.*"

 What do you mean by "due to larger differences at the southern high latitudes"? The sentence is not clear.
We changed the sentence to:
"*There is a pronounced seasonality in the differences in the SH, with the largest differences found during the austral summer (up to 4 %) and related to larger differences at the southern high latitudes.*"

 Could you report the numbers for "insignificant negative trends"? Do you refer to the P-value for "insignificant"?

For "insignificant", we refer to both the 2-sigma standard deviation and P-value. We changed the sentence to:

"*The time series of relative differences between IASI-A against UV-vis GB TOCs show insignificant negative drift in the NH (0.68±0.69 % decade$^{-1}$ and P-value= 0.05) and small negative trend in the SH (1.48±0.53% decade-1 and P-value=0.00), which satisfies the 1 – 3 % decade$^{-1}$ Ozone_cci requirements for stability of ozone measurements. Similar results are found with the IASI-A/FTIR TOC comparison.*"

Line 25 p. 14: The statement about the large biases found in the UTLS was already mentioned in Boynard et al. (2016), but still it is not fully understood. Could you address this question in your study?

We added this paragraph in the new IASI/FTIR section:

"*It should be noted that IASI is positively biased in the UTLS region, as reported in previous studies (e.g. Dufour et al., 2012; Gazeaux et al., 2013). Although Dufour et al. (2012) attempted to give some explanations for this particular feature, the exact reason for this overestimation is still not clear. One reason could be the use of inadequate a priori information. Note that FORLI uses only one single a priori profile (Hurtmans et al., 2012) that is the global mean profile of the McPeters/Labow/Logan climatology (McPeters et al., 2007). As shown by Bak et al. (2013), using tropopause-based ozone profile climatology can significantly improve the a priori. However, using dynamical a priori makes the comparison on a global scale less straightforward since a different a priori profile would be used at each IASI pixel*"

We also added the following paragraph in the summary:

"*Attempt of explanations for the larger bias found in the UTLS are given in Dufour et al. (2012) but no clear reason was found. A possible explanation could be the use of inadequate a priori information in that layer. The current version of FORLI uses as a priori profile a single global profile that is the mean of the McPeters/Labow/Logan climatology (McPeters et al., 2007). As shown by Bak et al. (2013), using tropopause-based ozone profile climatology can significantly improve the a priori. However, using dynamical a priori makes the comparison on a global scale less straightforward to analyze because the retrieval at each IASI pixel would be based on different a priori profiles.*"

---

## Referee Report (RR1)

*Review of*
*"Validation of the IASI FORLI/Eumetsat ozone products using satellite (GOME-2), ground-based (Brewer-Dobson, SAOZ) and ozonesonde measurements"*
*by Anne Boynard et al.*

The paper "Validation of the IASI FORLI/Eumetsat ozone products using satellite (GOME-2), ground-based (Brewer-Dobson, SAOZ) and ozonesonde measurements" by Anne Boynard et al validates the latest version of ozone data retrieved from spectra measured by IASI instrument using FORLI processing scheme, v20151001. The paper validates both total and partial ozone columns ozone data. The choice of reference instruments is logical and well thought: satellite-born GOME-2; ground-based Dobson, Brewer, SAOZ and FTIR, and ozonesondes. The dense data coverage of IASI instrument allows even to perform the initial drift assessment, using 9 years of the data.

The paper is very useful for users of IASI ozone data, even though the methods are quite identical to (Boyard et al, 2016). The paper under review is user-oriented, clearly written, and answers all main questions that a solid validation paper should answer : is there a bias, where is the bias concentrated, are the reasons for the bias known, how big are the uncertainties, how are the uncertainties geographically distributed, what are the major contributions to the uncertainty of the data, is there a drift, where is a drift, and are the reasons of the drift understood.

My recommendation is to publish the paper, subject to following minor changes:
1. The similarity of the methods to (Boynard 2016) is still quite misleading, despite the fact that compared to the previous version of the paper, the authors made an effort to make the changes between the two data versions transparent. The abstract and the summary should contain clear statements about changes (improvements) achieved in the new version obtained with new retrieval scheme with respect to the previous version.
2. The summary should mention if the authors recommend the IASI v20151001 ozone data for climatological studies. This is partly done in the answers to the reviews of the previous version of the paper, this should be included in the text of the paper.

Minor issues:
p.2, l. 8 : "In the troposphere, O3 plays different important roles …" Sounds strange, should be rephrased.
p. 15, l.27-29: "The standard deviation is maximum in the UTLS at Izaña and Lauder, which is due to strong O3 variability and large total retrieval error in this region as shown in Wespes et al. (2016). »
The Figure 4b in (Wespes et al 2016) indeed demonstrates that in tropical regions the estimated total retrieval error of vertical ozone profiles from IASI are larger than in middle latitudes, this indeed suggests that it would be the case for the ozone column as well. I would formulate it directly in this comment, rather than send the reader to the whole paper.

---

## Referee Report (RR2)

**Second Report of the manuscript titled "Validation of the IASI FORLI/Eumetsat ozone products using satellite (GOME-2), ground-based (Brewer-Dobson, SAOZ) and ozonesonde measurements"**

First of all, I would like to thank very much the authors for their very clear answers to the comments of the first report and for the very good quality of the updated manuscript. It is very much appreciated.

**There are still two points that I think should be addressed before publishing the present manuscript:**

1) In the end of the abstract, I would suggest to replace the sentence "However, since this difference in the drift values might be due only to the too short periods considered here associated with the high variability in TROPO O3 differences, a few more years are needed to confirm the observed negative drifts and evaluate them on the longer term"
by
"The observed negative drifts of IASI-A TROPO O3 product (8-16% decade-1) over 2008-2017 might be taken into consideration when deriving trends from this product and this time period."

Two reasons motivate this suggestion:
- Since the drift of IASI-A TROPO O3 in the northern hemisphere with ozonesondes for the time-period 2008-2017 is statistically significant, I am rather convinced that the 9 years of study is long enough. If 9 years is considered too short for a drift assessment, should it be considered too short for a TROPO O3 trend analysis? As it is specified in the text, you use 30 pairs of ozonesondes data per month throughout the northern hemisphere in order to assess the drift. Isn't it a good statistics of data?
Furthermore, same results are found with FTIR data for the 6 selected stations.
The significant negative drift for the time-period 2008-2017 seems to be real.
However, I would agree that the time-period 2011-2017 might be too short to show that the drift is decreasing and more years are needed indeed. I would suggest to make this point clearer.
In summary, I would suggest to clearly separate the conclusions found for 2008-2017 (real significant negative drift) from the conclusions found from 2011-2017.

- It is really important to inform the users about this clear negative trends over 2008-2017 and so to clearly state right in the abstract that it has to be taken into account for TROPO O3 trends analysis.

2) Figure 3 shows the relative difference between IASI-A and IASI-B for TOC. The period May 2013 – March 2015 seems to show rather a positive bias in the poles and in the tropics, while the period after September 2015 seems to show rather a negative bias. Could you explain how IASI-A TOC product measures $0.3 \pm 1.1$ % less ozone than IASI-B over the early period of time, while IASI-A TOC product gives $0.1 \pm 0.5$ % over the late period of time. I would rather expect a change in the sign.

How would you explain the positive bias in the tropics between May 2013 and March 2015?

**Minor comments:**
- You should choose between "Figure"and "Fig" when citing the figures
- Typos need to be fixed

---

## Author Response (AR2)

**Response to Referee #1**

**The authors would like to thank the referee for her/his general comments about the manuscript and her/his useful suggestions and corrections. Below are our responses to the comments brought up by the referee. The referee's comments and our responses are marked in blue and in black, respectively. In italic are the changes made in the manuscript.**

The paper "Validation of the IASI FORLI/Eumetsat ozone products using satellite (GOME-2), ground-based (Brewer-Dobson, SAOZ) and ozonesonde measurements" by Anne Boynard et al validates the latest version of ozone data retrieved from spectra measured by IASI instrument using FORLI processing scheme, v20151001. The paper validates both total and partial ozone columns ozone data. The choice of reference instruments is logical and well thought: satellite-born GOME-2; ground-based Dobson, Brewer, SAOZ and FTIR, and ozonesondes. The dense data coverage of IASI instrument allows even to perform the initial drift assessment, using 9 years of the data.

The paper is very useful for users of IASI ozone data, even though the methods are quite identical to (Boynard et al, 2016). The paper under review is user-oriented, clearly written, and answers all main questions that a solid validation paper should answer : is there a bias, where is the bias concentrated, are the reasons for the bias known, how big are the uncertainties, how are the uncertainties geographically distributed, what are the major contributions to the uncertainty of the data, is there a drift, where is a drift, and are the reasons of the drift understood.

My recommendation is to publish the paper, subject to following minor changes:

1. The similarity of the methods to (Boynard 2016) is still quite misleading, despite the fact that compared to the previous version of the paper, the authors made an effort to make the changes between the two data versions transparent. The abstract and the summary should contain clear statements about changes (improvements) achieved in the new version obtained with new retrieval scheme with respect to the previous version.

As suggested by the reviewer, we made the following changes:

Summary Section:
*"Compared to the previous version of FORLI-O3 (v20140922), several improvements were introduced in FORLI-O3 v20151001, including absorbance look-up tables recalculated to cover a larger spectral range using the 2012 HITRAN spectroscopic database (Rothman et al., 2013), with additional numerical corrections. This leads to a change of ~4% in the Total Ozone Column (TOC) product, which is mainly associated with a decrease in the retrieved O3 concentration in the middle stratosphere (above 30 hPa/25 km)."*

Abstract:
*"Compared to the previous version of FORLI-O3 (v20140922), several improvements were introduced in FORLI-O3 v20151001, including absorbance look-up tables recalculated to cover a larger spectral range, with additional numerical corrections. This leads to a change of ~4% in the Total Ozone Column (TOC) product, which is mainly associated with a decrease in the retrieved O3 concentration in the middle stratosphere (above 30 hPa/25 km)."*

2. The summary should mention if the authors recommend the IASI v20151001 ozone data for climatological studies. This is partly done in the answers to the reviews of the previous version of the paper, this should be included in the text of the paper.

We added the following paragraph in the summary:
*"The IASI-A TOC relative differences against independent measurements showed small or insignificant negative*

*decadal drifts for the period 2008-2017, which indicates that the current IASI-A TOC products are homogeneous and reliable for trend studies. The IASI-A TROPO O₃ relative differences against sonde and FTIR data showed significant negative drifts for the period 2008-2017. It is therefore recommended for trend studies to wait for the new homogeneous IASI climate time series, which will be reprocessed using the European Centre for Medium-Range Weather Forecast (ECMWF) ERA5 temperatures reanalysis (Hersbach and Dee, 2016) and reprocessed IASI Level-1 data."*

Hersbach, H. and Dee, D.: ERA5 reanalysis is in production, ECMWF Newsletter, p. 7, 2016.

Minor issues:
p.2, l. 8 : "In the troposphere, O3 plays different important roles …" Sounds strange, should be rephrased.
We removed the word "important".

p. 15, l.27-29: "The standard deviation is maximum in the UTLS at Izaña and Lauder, which is due to strong O3 variability and large total retrieval error in this region as shown in Wespes et al. (2016). »
The Figure 4b in (Wespes et al 2016) indeed demonstrates that in tropical regions the estimated total retrieval error of vertical ozone profiles from IASI are larger than in middle latitudes, this indeed suggests that it would be the case for the ozone column as well. I would formulate it directly in this comment, rather than send the reader to the whole paper.
As suggested by the referee we added this sentence in the manuscript:
*"Indeed their Fig. 4b demonstrated that in tropical regions the estimated total retrieval error of vertical ozone profiles from IASI are larger than in middle latitudes, which suggests that it would be the case for the ozone column as well."*

**Response to Referee #2**

**The authors would like to thank the referee for her/his second review. Below are our responses to the comments brought up by the referee. Referee's comments and our replies are marked in blue and in black, respectively. In italic are the changes made in the manuscript.**

There are still two points that I think should be addressed before publishing the present manuscript:
1) In the end of the abstract, I would suggest to replace the sentence "However, since this difference in the drift values might be due only to the too short periods considered here associated with the high variability in TROPO O3 differences, a few more years are needed to confirm the observed negative drifts and evaluate them on the longer term"
by
"The observed negative drifts of IASI-A TROPO O3 product (8-16% decade-1) over 2008-2017 might be taken into consideration when deriving trends from this product and this time period."
Two reasons motivate this suggestion:
- Since the drift of IASI-A TROPO O3 in the northern hemisphere with ozonesondes for the time-period 2008-2017 is statistically significant, I am rather convinced that the 9 years of study is long enough. If 9 years is considered too short for a drift assessment, should it be considered too short for a TROPO O3 trend analysis? As it is specified in the text, you use 30 pairs of ozonesondes data per month throughout the northern hemisphere in order to assess the drift. Isn't it a good statistics of data?
Furthermore, same results are found with FTIR data for the 6 selected stations.
The significant negative drift for the time-period 2008-2017 seems to be real.
However, I would agree that the time-period 2011-2017 might be too short to show that the drift is decreasing and more years are needed indeed. I would suggest to make this point clearer.
In summary, I would suggest to clearly separate the conclusions found for 2008-2017 (real significant negative drift) from the conclusions found from 2011-2017.
- It is really important to inform the users about this clear negative trends over 2008-2017 and so to clearly state right in the abstract that it has to be taken into account for TROPO O3 trends analysis.

As suggested by the reviewer we changed
"However, since this difference in the drift values might be due only to the too short periods considered here associated with the high variability in TROPO O3 differences, a few more years are needed to confirm the observed negative drifts and evaluate them on the longer term"

by
"*The observed negative drifts of IASI-A TROPO O3 product (8-16% decade-1) over 2008-2017 might be taken into consideration when deriving trends from this product and this time period.*"

2) Figure 3 shows the relative difference between IASI-A and IASI-B for TOC. The period May 2013 – March 2015 seems to show rather a positive bias in the poles and in the tropics, while the period after September 2015 seems to show rather a negative bias. Could you explain how IASI-A TOC product measures 0.3 ± 1.1 % less ozone than IASI-B over the early period of time, while IASI-A TOC product gives 0.1 ± 0.5 % over the late period of time. I would rather expect a change in the sign.
How would you explain the positive bias in the tropics between May 2013 and March 2015?

The quoted values are very small, lower than 0.5% and their standard deviation exceeds the mean values. Also, over the period before March 2015, differences can reach -14% while over the period after October 2015 the lowest difference is -4% (the color scale is saturated). This is reflected in the standard deviation values (1.1% versus 0.5%). Moreover, it is not that easy to expect a change in the sign since the step in the differences is 0.2%.

Between May 2013 and March 2015, the bias is mainly positive in the tropics but it is less than 0.2%. It should be noted that the locations of the observations differs and that there is a documented and known small radiance bias between the two instruments (about 0.5K). This will be the subject of another paper, which our observations of the small mismatch in ozone triggered.

Minor comments:
- You should choose between "Figure"and "Fig" when citing the figures
We followed AMT journal guidelines indicated at https://www.atmospheric-measurement-techniques.net/for_authors/manuscript_preparation.html:
The abbreviation "Fig." should be used when it appears in running text and should be followed by a number unless it comes at the beginning of a sentence, e.g.: "The results are depicted in Fig. 5. Figure 9 reveals that...".

- Typos need to be fixed
This is usually fixed during the copy-editing phase.

[revised manuscript text omitted]